# Cell-type-specific *cis*-regulatory divergence in gene expression and chromatin accessibility revealed by human-chimpanzee hybrid cells

**Ban Wang†, Alexander L Starr†, Hunter B Fraser\***

Department of Biology, Stanford University, Stanford, United States

**\*For correspondence:**
hbfraser@stanford.edu

†These authors contributed equally to this work

**Competing interest:** The authors declare that no competing interests exist.

**Abstract** Although gene expression divergence has long been postulated to be the primary driver of human evolution, identifying the genes and genetic variants underlying uniquely human traits has proven to be quite challenging. Theory suggests that cell-type-specific *cis*-regulatory variants may fuel evolutionary adaptation due to the specificity of their effects. These variants can precisely tune the expression of a single gene in a single cell-type, avoiding the potentially dele-terious consequences of *trans*-acting changes and non-cell type-specific changes that can impact many genes and cell types, respectively. It has recently become possible to quantify human-specific *cis*-acting regulatory divergence by measuring allele-specific expression in human-chimpanzee hybrid cells—the product of fusing induced pluripotent stem (iPS) cells of each species *in vitro*. However, these *cis*-regulatory changes have only been explored in a limited number of cell types. Here, we quantify human-chimpanzee *cis*-regulatory divergence in gene expression and chromatin accessibility across six cell types, enabling the identification of highly cell-type-specific *cis*-regulatory changes. We find that cell-type-specific genes and regulatory elements evolve faster than those shared across cell types, suggesting an important role for genes with cell-type-specific expression in human evolution. Furthermore, we identify several instances of lineage-specific natural selection that may have played key roles in specific cell types, such as coordinated changes in the *cis*-regulation of dozens of genes involved in neuronal firing in motor neurons. Finally, using novel metrics and a machine learning model, we identify genetic variants that likely alter chromatin accessibility and transcription factor binding, leading to neuron-specific changes in the expression of the neurode-velopmentally important genes *FABP7* and *GAD1*. Overall, our results demonstrate that integrative analysis of *cis*-regulatory divergence in chromatin accessibility and gene expression across cell types is a promising approach to identify the specific genes and genetic variants that make us human.

## eLife assessment

This is an **important** study that leverages a human-chimpanzee tetraploid iPSC model to test whether *cis*-regulatory divergence between species tends to be cell type-specific. The evidence supporting the study's primary conclusions together provide **convincing** evidence for enrichment of species differences in gene regulation in cell type-specific genes and regulatory elements, moti-vating future work with larger sample sizes of cell lines. This work will be of broad interest in evolu-tionary and functional genomics.

## Introduction

In the past few million years, humans have evolved a multitude of unique phenotypes (*Shave et al., 2019*; *Vanderhaeghen and Polleux, 2023*). For example, our cardiovascular system has evolved to enable extended periods of physical exertion and the unique aspects of our nervous system enable human language and toolmaking (*Shave et al., 2019*; *Vanderhaeghen and Polleux, 2023*). Previous research suggests that much of human adaptation may be caused by changes in gene expression (*Fraser, 2013*; *Kelley and Wilson, 2020*; *Reilly and Noonan, 2016*; *Romero et al., 2012*). To catalog these changes, studies have compared gene expression in post-mortem tissues of humans and our closest living relatives, chimpanzees (*Blake et al., 2020*; *Kelley and Gilad, 2020*; *Ma et al., 2022*). Although thousands of differentially expressed genes have been identified in post-mortem samples, it is generally not possible to disentangle the effects of genetic differences from the effects of confounding factors such as differences in diet, environment, cell type abundances, age, post-mortem interval, etc. In addition, for many traits the relevant gene expression differences may be specific to early development, but it is impossible to study fetal development *in vivo* in non-human great apes due to both ethical and technical difficulties. To circumvent these issues, several groups have used great ape iPS cells to study differences in gene expression in cell types present in early development (*Benito-Kwiecinski et al., 2021*; *Field et al., 2019*; *Kanton et al., 2019*; *Pavlovic et al., 2018*). While the use of iPS cells addresses many of the confounding factors present in post-mortem comparisons, they also introduce new issues such as interspecies differences in iPS cell differentiation kinetics, efficiency, and maturation. Overall, it remains tremendously challenging to identify human-specific changes in gene expression, which limits our ability to link expression differences to either phenotypic differences or natural selection in the human lineage.

One particularly powerful means of studying the evolution of gene expression is through the measurement of allele specific expression (ASE) in hybrids between two species (*Combs et al., 2018*; *Fraser, 2011*; *Hu et al., 2022*; *Mack and Nachman, 2017*; *Wittkopp and Kalay, 2011*). This approach has the advantage of eliminating many confounding factors inherent to interspecies comparisons, including differences in cell type composition, environmental factors, developmental stage, and response to differentiation protocols. Because the trans-acting environments of the two alleles in a hybrid are identical, ASE has the additional benefit of reflecting only *cis*-regulatory changes, which are thought to be less pleiotropic and more likely to drive evolutionary adaptation than broader *trans*-acting changes (*Agoglia et al., 2021*; *Gokhman et al., 2021*; *Prud'homme et al., 2007*; *Wittkopp and Kalay, 2011*). Furthermore, ASE enables the use of powerful methods that can detect lineage-specific natural selection and, as a result, contribute to our understanding of the selective pressures that have shaped the evolution of a wide variety of species (*Fraser, 2011*). Until recently, it has not been possible to disentangle *cis*- and *trans*-acting changes fixed in the human lineage since humans cannot hybridize with any other species. However, the development of human-chimpanzee hybrid iPS cells via *in vitro* cell fusion enables measurement of ASE in a wide variety of tissues and developmental contexts (*Agoglia et al., 2021*; *Gokhman et al., 2021*; *Song et al., 2021*). This provides an effective platform to investigate general principles of hominid gene expression evolution, detect lineage-specific selection, and identify candidate gene expression changes underlying human-specific traits.

While gene expression divergence between humans and chimpanzees is well-studied, there has been less focus on epigenetic differences, many of which are likely to underlie divergent gene expression (*García-Pérez et al., 2021*; *Kozlenkov et al., 2020*; *Vermunt et al., 2016*; *Trizzino et al., 2017*). Furthermore, these studies, regardless of whether they utilize postmortem tissues or cell lines, are subject to the same confounding factors mentioned above. Analogous to ASE, one can use the assay for transposase accessible chromatin using sequencing (ATAC-seq) in interspecies hybrids to measure allele-specific chromatin accessibility (ASCA; *Buenrostro et al., 2013*; *Corces et al., 2017*; *Liang et al., 2021*; *Zhang et al., 2020*). As with ASE, ASCA is unaffected by many confounders inherent to between-species comparisons and only measures *cis*-regulatory divergence. Perhaps most importantly, ASCA can implicate specific regulatory elements that likely underlie gene expression differences. These regulatory elements can then be more closely studied to identify the likely causal genetic variants and the molecular mechanisms by which those variants alter gene expression.

Here, we generated RNA-seq and ATAC-seq data from six cell types, derived from human-chimpanzee hybrid iPS cells, and quantified ASE and ASCA. Using this dataset, we identified thousands of genes and *cis*-regulatory elements showing cell-type-specific ASE and ASCA. We found

that cell-type-specific genes and *cis*-regulatory elements are more likely to have divergent expression and accessibility than their more broadly expressed/accessible counterparts. Furthermore, we provide evidence for polygenic selection on the expression level of genes associated with physiologically relevant gene sets including sodium channels and syntaxin-binding proteins in motor neurons. Finally, we use newly developed metrics and machine learning algorithms to link cell-type-specific differences in chromatin accessibility and gene expression and identify putative causal mutations underlying these differences. Using this pipeline we identified motor neuron-specific increases in promoter chromatin accessibility and gene expression for *FABP7*, which plays a key role in neurodevelopment but is not well-studied in neurons. In addition, we focus on a human-accelerated region (HAR) near the promoter of *GAD1*. While this region is accessible in all cell types, both the accessibility of the HAR and the expression of *GAD1* are only chimpanzee-biased in motor neurons. Analysis of scRNA-seq from human and chimpanzee brain organoids showed that increased expression of *GAD1* also occurs in ventral forebrain inhibitory neurons. Overall, this study provides insight into the evolution of gene expression in hominids as well as a resource that will inform functional genomic dissection of human-specific traits.

## Results

### *Cis*-regulatory divergence of gene expression in six cell types is largely cell-type-specific or shared across all cell types

To measure genome-wide *cis*-regulatory divergence in gene expression, we performed RNA-seq on six cell types derived from human-chimpanzee hybrid iPS cells (*Figure 1a*). The cell types profiled were from six diverse developmental lineages including the motor neuron (MN), cardiomyocyte (CM), hepatocyte progenitor (HP), pancreatic progenitor (PP), skeletal myocyte (SKM), and retinal pigment epithelium (RPE) lineages. These represent all three germ layers and a variety of organs (*Figure 1a*). It is worth noting that these differentiations do not necessarily lead to a pure population of cells, but rather a population of cells with different levels of maturity along a particular developmental lineage. For example, the SKM population likely contains fully differentiated muscle fibers as well as a small population of proliferating satellite cells; for clarity we refer to this as the SKM cell type. As these different cell types are not shared between tissues, we use cell-type-specific and tissue-specific interchangeably throughout the manuscript.

Two independently generated hybrid lines were differentiated for each cell type and at least two biological replicates per hybrid line per cell type were collected (see Methods). Each cell type was sequenced to an average depth of 134 million paired-end reads (*Figure 1—figure supplement 1*). We used a computational pipeline to quantify ASE adapted from the pipeline introduced by *Agoglia et al., 2021*. Briefly, we computed ASE by mapping reads to both the human and chimpanzee genomes, correcting for mapping bias, and assigning reads to the human or chimpanzee genome if a read contained one or more human-chimpanzee single nucleotide differences (see Methods).

As expected, the samples clustered predominantly by cell type (*Figure 1b–c*). Within four of the six cell type clusters, individual samples clustered by line rather than species of origin, potentially indicating line to line variability in differentiation (*Figure 1b*). This highlights the importance of measuring ASE which, by definition, is measured within each line and so is robust to variability between lines. Indeed, when performing PCA within cell types using allelic counts (i.e. counting reads from the human allele and chimpanzee allele separately), human and chimpanzee species differences were clearly separated by principal component (PC) 1 or PC2 in each cell type (*Figure 1d*, *Figure 1—figure supplement 2*). To assess the success of our differentiations, we examined each cell type for the expression of known marker genes in our RNA-seq data (*Figure 1e*, *Figure 1—figure supplement 3*). All cell types express canonical marker genes and do not express pluripotency markers, indicating that the differentiations were successful (*Figure 1e*, see *Figure 1—figure supplement 3* for an extensive discussion of the differentiations; *Burridge et al., 2014*; *Chal et al., 2016*; *Korytnikov and Nostro, 2016*; *Mallanna and Duncan, 2013*; *Maury et al., 2015*; *Sharma et al., 2019*).

Because our hybrid cells were grown concurrently with their human and chimpanzee diploid 'parental' cells, we performed an additional check for purity of the hybrid lines by quantifying genome-wide ASE. We noticed that among our 25 RNA-seq samples, the two PP hybrid2 samples had a slight bias towards higher expression from the chimpanzee alleles across all chromosomes. This is likely

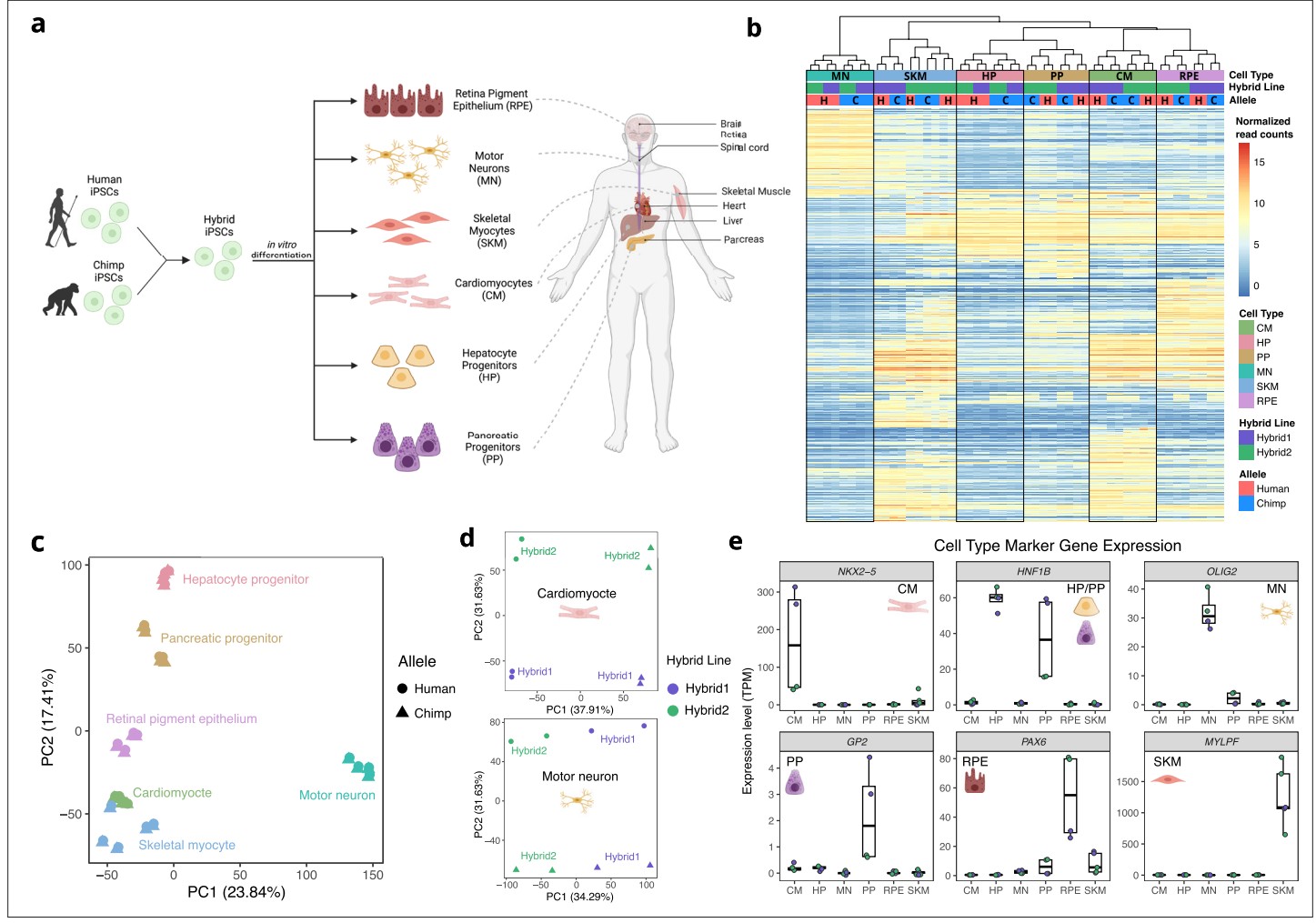

**Figure 1.** Allele-specific expression across diverse human-chimpanzee hybrid cell types. (**a**) Six cell types were differentiated from human-chimpanzee hybrid induced pluripotent stem cells. These six cell types represent diverse body systems, including motor neurons for the central nervous system, retinal pigment epithelium for eye, skeletal myocytes for skeletal muscle, cardiomyocytes for the heart, hepatocyte progenitors for the liver, and pancreatic progenitors for the pancreas. (**b**) Heatmap showing the result of hierarchical clustering performed on genes with highly variable normalized allele counts. (**c**) Result of running PCA on normalized allelic counts for all samples and cell types. (**d**) Result of PCA performed on normalized allele counts for each individual cell type separately. Cardiomyocytes and motor neurons are shown here. (**e**) Expression of marker genes for each cell type.

The online version of this article includes the following figure supplement(s) for figure 1:

**Figure supplement 1.** Sequencing depth across samples for the RNA-seq data.

**Figure supplement 2.** PCA on allelic counts from RNA-seq for individual cell types.

**Figure supplement 3.** Marker gene expression in different cell types.

**Figure supplement 4.** Simulated chimpanzee parental contamination of hybrid RNA-seq data and correction.

due to a small fraction of contaminating chimpanzee cells in these samples. We corrected for this by reducing the chimpanzee allele counts such that the number of reads assigned to the human and chimpanzee alleles was equal. By simulating contamination of a hybrid sample with chimpanzee cells, we found that this correction was conservative and that the log fold-change estimates were largely unaffected by contamination after this correction (Methods, *Figure 1—figure supplement 4*).

We next investigated which genes were differentially expressed between the human and chimpanzee allele in each cell type (*Figure 2a*). We identified thousands of genes showing significantly biased ASE in each cell type at a false discovery rate (FDR) cutoff of 0.05 (*Figure 2b*). We detected

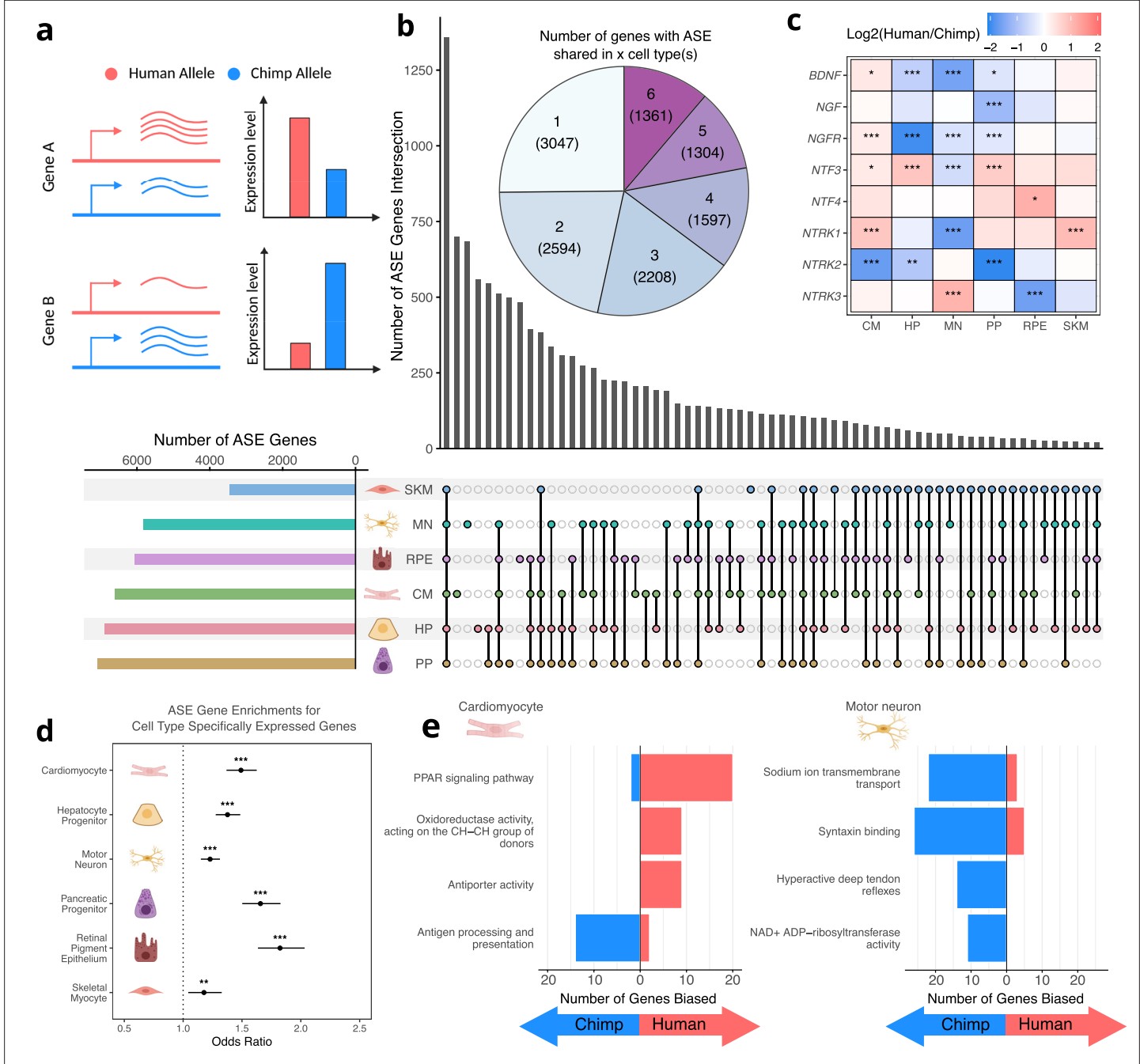

**Figure 2.** Human-chimpanzee ASE is largely cell-type-specific. (**a**) Outline of measurement of allele-specific expression. Reads from the human and chimpanzee alleles are counted and differences in read counts identified. (**b**) Thousands of genes with ASE were identified for each cell type, with many genes only showing ASE in a single-cell type (***Conway et al., 2017***). (**c**) The neurotrophins and their receptors as examples of genes showing cell-type-specific ASE patterns. DESeq2 estimate of log₂ fold-change (human/chimpanzee) are shown in the heatmap and significance is indicated by asterisks where *** indicates FDR <0.005, ** indicates FDR <0.01, and * indicates FDR <0.05. Zero asterisks (i.e. a blank box) indicates FDR >0.05. (**d**) Plot showing that genes with ASE are enriched for genes showing cell-type-specific expression patterns across all cell types. Asterisks indicate p-values rather than FDR using the same system as in 2 c. (**e**) Top gene sets with evidence for lineage-specific selection in cardiomyocytes and motor neurons are shown. The length of the bars indicates the number of genes in a category with biased expression in each cell type.

The online version of this article includes the following figure supplement(s) for figure 2:

**Figure supplement 1.** *NTF3*, *NTRK1,* and *NTRK3* have cell-type-specific allele-specific expression.

*Figure 2 continued on next page*

*Figure 2 continued*

**Figure supplement 2.** Genes expressed in only one cell type are enriched for genes with ASE.

**Figure supplement 3.** Cell type-specifically expressed genes are enriched for genes with ASE across FDR cutoffs.

**Figure supplement 4.** Cell type-specifically expressed genes are enriched for genes with ASE regardless of expression level.

**Figure supplement 5.** Cell type-specifically expressed genes are enriched for genes with ASE when different samples are used to identify cell-type-specific and ASE genes.

**Figure supplement 6.** Controlling for ASE variance generally does not eliminate enrichment of ASE genes in cell type-specifically expressed genes.

**Figure supplement 7.** Controlling for probability of haploinsufficiency (pHI) generally does not eliminate enrichment of ASE genes in cell type-specifically expressed genes.

**Figure supplement 8.** Validation of increased *NR1H3* expression in the human lineage.

**Figure supplement 9.** ASE and differential expression of *SCN1B*, *SCN2B*, and *SYT2* across cell types.

a comparable number of ASE genes in all cell types except SKM. As a result, we repeated all subsequent analyses both including and excluding SKM and obtained qualitatively similar results regardless of whether SKM was included.

While a considerable number of genes had significant ASE in all cell types, many more genes only had significant ASE in a single cell type, suggesting cell-type-specific *cis*-regulatory divergence (*Figure 2b*). A notable family of developmentally important genes that exemplifies differences in ASE across cell types is the neurotrophins and their receptors (*Figure 2c*; *Caporali and Emanueli, 2009*; *Huang and Reichardt, 2001*). For example, *NTRK3*, which plays a key role in the development of the nervous system, is only differentially expressed in RPE and MN but is chimpanzee-biased in RPE and human-biased in MN (*Figure 2—figure supplement 1*; *Ichim et al., 2012*; *Naito et al., 2017*). In addition, the gene coding for its primary ligand (*NTF3*) is differentially expressed in a variety of cell types yet is human-biased in all cell types except MN in which it is chimpanzee-biased (*Figure 2—figure supplement 1*). *NTRK1* differential expression is similarly tissue-specific as it is strongly chimpanzee-biased in MN, but human-biased in CM and SKM (*Figure 2—figure supplement 1*). These results indicate that the regulatory landscape of these genes has undergone many complex *cis*-regulatory changes as the human and chimpanzee lineages have diverged.

To further investigate the relationship between tissue-specificity and ASE, we asked whether genes with variable expression across tissues are more likely to show ASE. Using a standard definition of cell-type-specific genes—those with detectable expression in only one cell type in our study—we found that cell-type-specific genes were typically enriched for ASE in the one cell type where they are expressed (*Figure 2—figure supplement 2*; *GTEx Consortium, 2017*; *Jain and Tuteja, 2019*). However, other cell-type-specific expression patterns such as uniquely low expression in a particular cell type may also indicate an important dosage sensitive function in that cell type. We therefore focused on a broader definition of cell type-specificity in which genes that are differentially expressed between one cell type and all others in our study (FDR < 0.05 for each pairwise comparison) are considered cell-type-specific for that cell type. We found that this more inclusive definition, which identified many more cell-type-specific genes, showed an even more significant ASE enrichment than the narrower definition (*Figure 2d*). This result is not sensitive to the choice of FDR cutoff (*Figure 2—figure supplement 3*) nor driven solely by a subgroup of highly expressed genes (*Figure 2—figure supplement 4*). This trend is also robust to separating samples into two groups and using one to define cell-type-specific genes and the other to identify differentially expressed genes. This controls for spurious relationships that can result when the same data are used to define two different quantities which are then compared (*Figure 2—figure supplement 5*; *Fraser, 2019*).

This enrichment (*Figure 2d*) suggests that tissue-specific genes may have less constraint and/or more frequent positive selection on their expression. We reasoned that if the trend was solely driven by constraint, then controlling for constraint—even if imperfectly—would be expected to reduce the strength of the relationship. To investigate this, we binned genes by their variance in ASE across a large cohort of human samples which we have previously shown acts as a reasonable proxy for evolutionary constraint on gene expression (*Castel et al., 2020*; *Starr et al., 2023*). Across cell types, we generally observe significant enrichments in each bin and little difference in enrichment between bins, suggesting that differences in constraint on expression of cell-type-specific vs. ubiquitously expressed genes are not solely responsible for our observations (*Figure 2—figure supplement 6*). Furthermore,

we observe even stronger enrichments using an alternative constraint metric, the probability of haplo-insufficiency score (pHI) likely due to the larger number of genes for which pHI can be calculated (*Figure 2—figure supplement 7*; *Collins et al., 2022*). Overall, our analysis suggests that differences in constraint are unlikely to fully explain these trends, suggesting a potential role for positive selection.

## Lineage-specific selection has acted on tissue-specific gene expression divergence

Next, we sought to use our RNA-seq data to identify instances of lineage-specific selection. In the absence of positive selection, one would expect that an approximately equal number of genes in a pathway would have human-biased vs. chimpanzee-biased ASE. Significant deviation from this expectation (as determined by the binomial test) rejects the null hypothesis of neutral evolution, instead providing evidence of lineage-specific selection on this pathway. Using our previously published modification of this test that incorporates a tissue-specific measure of constraint on gene expression, we detected several signals of lineage-specific selection, some of which were cell-type-specific (*Starr et al., 2023*, *Supplementary file 1*). Notably, the four most significant enrichments were specific to motor neurons and cardiomyocytes and are highly relevant to those cell types (*Figure 2e*; *Supplementary file 1*). In cardiomyocytes, the top pathway was 'PPAR signaling pathway' which plays a key role in the regulation of heart morphology and lipid metabolism (*Montaigne et al., 2021*). For example, *NR1H3* (also known as *LXRA*) is strongly upregulated in human cardiomyocytes as well as all other cell types (*Figure 2—figure supplement 8a*). Furthermore, this upregulation appears to have occurred in the human lineage based on data from non-hybrid cardiomyocytes as well as adult hearts (*Figure 2—figure supplement 8b*; *Blake et al., 2020*; *Pavlovic et al., 2018*). Hybrid cells are essential in determining that the human-specific upregulation of *NR1H3* in cardiomyocytes has a strong genetic component as *NR1H3* expression is very responsive to diet and other environmental factors (*Wang and Tontonoz, 2018*).

In motor neurons, multiple categories showed a strong bias toward higher chimpanzee expression including 'sodium ion transmembrane transport' and 'syntaxin binding'. The genes in these categories are of fundamental importance to the function of motor neurons as sodium ion transporters control excitability and syntaxin binding proteins control the release of neurotransmitters from synaptic vesicles (*Brose et al., 2019*; *Meisler et al., 2021*). Interestingly, several key genes in these sets appear to have human-chimpanzee differences in expression that extend beyond motor neurons to other neuronal types. For example, *SCN1B*, *SCN2B*, and *SYT2* have chimpanzee-biased ASE in our MN data. In contrast, these genes have been observed to have lower expression in human cortical glutamatergic neurons compared to neurons from chimpanzees and rhesus macaques (*Figure 2—figure supplement 9*; *Kozlenkov et al., 2020*). We note that several other genes in these genes sets are not differentially expressed between humans and chimpanzees in this cortical neuron dataset, emphasizing the importance of studying individual neuron types (*Kozlenkov et al., 2020*). Overall, the strong bias in gene expression of sodium ion transporters and syntaxin binding proteins we observe suggests lineage-specific selection that may have altered the electrophysiological properties of human motor neurons.

## Patterns of allele-specific chromatin accessibility reveal divergent *cis*-regulatory elements

While ASE provides insight into what gene expression changes might underlie phenotypic differences between humans and chimpanzees, in the absence of additional data it is very difficult to prioritize which specific mutations might cause expression divergence. To begin to fill this gap, we generated ATAC-seq data from five of the six cell types (all except RPE), with each cell type sequenced to an average depth of 184 million paired-end reads (*Figure 3—figure supplement 1*). ATAC-seq uses a hyperactive Tn5 transposase to cleave DNA that is not bound by nucleosomes to enrich for accessible chromatin (*Figure 3a*), a hallmark of active *cis*-regulatory elements (CREs) (*Buenrostro et al., 2013*; *Corces et al., 2017*). We estimated ASCA for individual open chromatin peaks by mapping reads to both species' reference genomes, correcting for mapping bias, generating a unified list of peaks across all samples, and then counting reads supporting each allele in each peak (see Methods). After extensive filtering, we were left with 73,360 peaks across the five cell types with many fewer retained in SKM due to lower sequencing depth (*Figure 3—figure supplement 2*). As expected, most genes

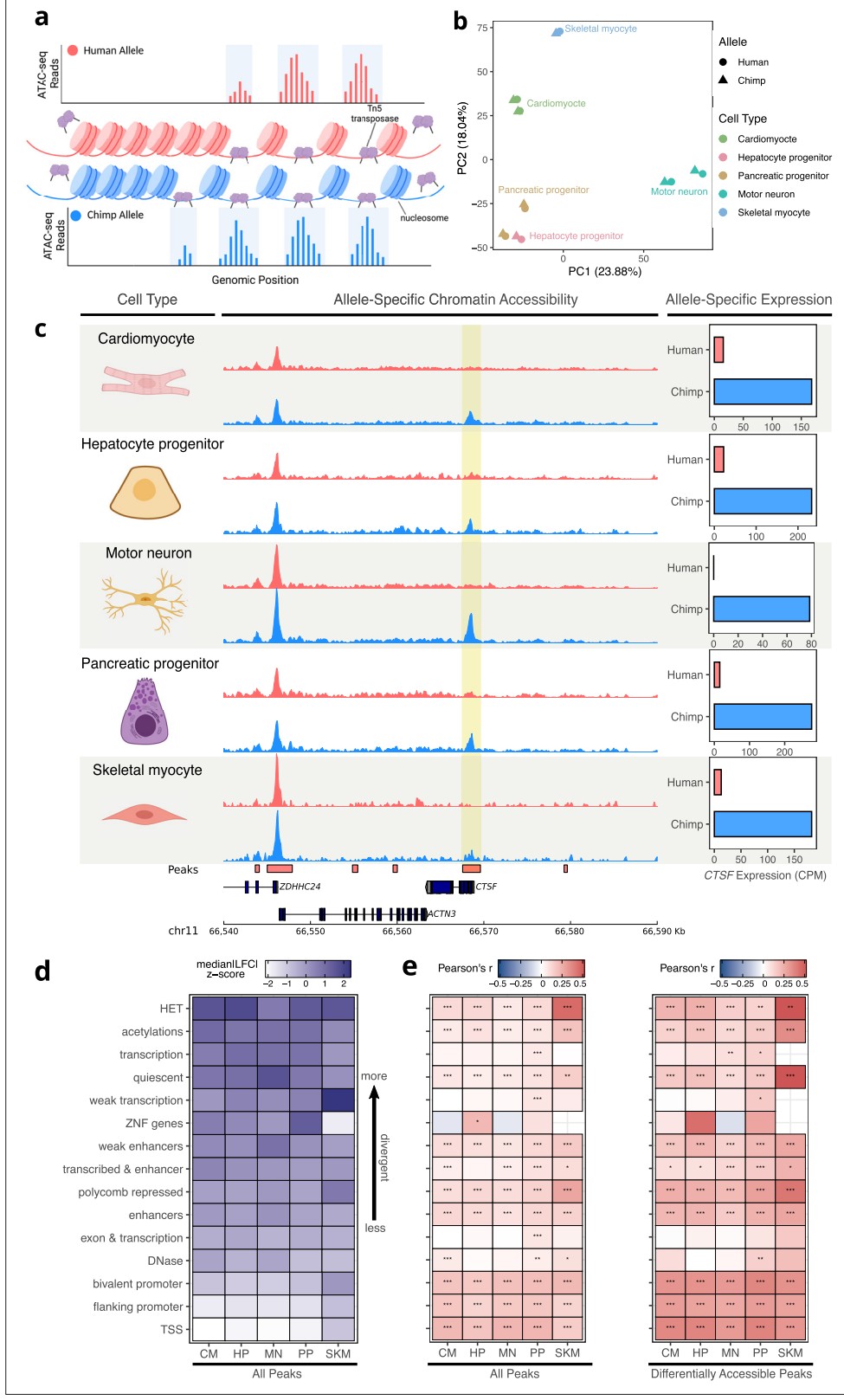

**Figure 3.** Allele-specific chromatin accessibility across diverse human-chimpanzee hybrid cell types. (**a**) Schematic outlining the ATAC-seq protocol. A hyperactive transposase cleaves accessible DNA and adds adapters enabling measurement of chromatin accessibility. (**b**) PCA on normalized allelic counts from ATAC-seq. (**c**) ASCA in the promoter of *CTSF,* and ASE for the *CTSF* gene. (**d**) Differences in ASCA were quantified and plotted separately

*Figure 3 continued*

based on chromHMM annotation. The order is based on the median of z-score transformed absolute log fold-change between human and chimpanzee across all cell types, with higher z-scores indicating greater divergence in accessibility. (**e**) Pearson correlation between ASE and ASCA for all cell types with all peaks (left) or only differentially accessible peaks (right, defined as peaks with nominal binomial p-value less than 0.05). Pearson's r values are shown in the heatmap and significance is indicated by asterisks where *** indicates p<0.005, ** indicates p<0.01, and * indicates p<0.05. Zero asterisks (i.e. a blank box) indicates p>0.05.

The online version of this article includes the following figure supplement(s) for figure 3:

**Figure supplement 1.** Sequencing depth in the ATAC-seq dataset.

**Figure supplement 2.** Number of CREs identified in each cell type.

**Figure supplement 3.** Number of CREs per gene identified in each cell type.

**Figure supplement 4.** Heatmap and clustering of ATAC-seq samples.

**Figure supplement 5.** PCA of HP and PP ATAC-seq samples only.

**Figure supplement 6.** PCA separates the human and chimpanzee alleles in the ATAC-seq data.

**Figure supplement 7.** ASCA predicts ASE across a variety of epigenetic states.

**Figure supplement 8.** ASE and ASCA correlations for TSS-proximal and distal CREs.

**Figure supplement 9.** ASE and ASCA correlation for cell-type-specific and ubiquitously expressed genes.

**Figure supplement 10.** Down-sampling eliminates sequencing depth-induced bias in the number of peaks called in each cell type.

**Figure supplement 11.** Cell-type-specific ATAC-seq peaks are enriched for ASCA across log fold-change cutoffs.

**Figure supplement 12.** Cell-type-specific ATAC-seq peaks are generally enriched for ASCA across p-value cutoffs.

**Figure supplement 13.** Cell-type-specific ATAC peaks are enriched for ASCA using a broader definition of cell-type-specific.

**Figure supplement 14.** Cell-type-specific ATAC peaks are enriched for ASCA using a broader definition of cell-type-specific.

**Figure supplement 15.** Cell-type-specific ATAC peaks are generally enriched for ASCA regardless of sequence constraint.

**Figure supplement 16.** Relationship between number of SNPs and $-\log_{10}$(FDR) in (**a**) ASE and $-\log_{10}$(p-value) (**b**) ASCA.

**Figure supplement 17.** Relationship between number of SNPs and absolute $\log_2$ fold-change in (**a**) ASE and (**b**) ASCA.

**Figure supplement 18.** Cell type-specifically expressed genes are enriched for genes with ASE when stratifying by the number of SNPs per gene.

**Figure supplement 19.** Cell-type-specific peaks are enriched for ASCA when stratifying by the number of SNPs per peak.

**Figure supplement 20.** Distribution of dEE across cell types.

**Figure supplement 21.** Distribution of dCAE across cell types.

---

had only a few peaks assigned to them (*Figure 3—figure supplement 3*). Cell types clustered well using PC1 and PC2 of the ATAC-seq data, except for the HP sample which clustered closely with PP samples (*Figure 3b*, *Figure 3—figure supplement 4*). However, performing PCA on just the HP and PP samples clearly separates the two cell types (*Figure 3—figure supplement 5*). Within each cell type, species differences were clearly separated by PC1 or PC2 (*Figure 3—figure supplement 6*). As an example of ASCA, the accessibility of the promoter of *CTSF* was strongly chimpanzee-biased, mirroring the chimpanzee-biased ASE for this gene (*Figure 3c*).

As a first step in analyzing the ATAC-seq data, we intersected the peaks we identified with the genomic annotations of chromatin states. These fifteen categories, predicted across many tissues and cell types by the chromHMM model (*Vu and Ernst, 2022*), include terms such as 'TSS' (transcription start site) and 'enhancer'. We then plotted the median of the absolute human-chimpanzee ASCA log fold-changes for each chromatin state and cell type (*Figure 3d*). The TSS and promoter

annotations were the least divergent in their accessibility, whereas regions of heterochromatin were the most divergent (*Figure 3d*). To explore the relationship between interspecies differences in ASCA and ASE, we assigned peaks to the nearest TSS and computed the Pearson correlation between the allelic log fold-change of chromatin accessibility and expression within each cell type and chromatin state. As expected, TSS and promoter annotations showed the strongest correlation between ASCA and ASE, and correlations were stronger when including only differentially accessible peaks (*Figure 3e*, Methods). Intriguingly however, ASCA of regions annotated as heterochromatin, polycomb repressed, or quiescent were as strongly correlated with ASE as elements identified as enhancers or DNase hypersensitivity sites (*Figure 3e*). Notably this result is robust to how peaks and chromHMM annotations were intersected (*Figure 3—figure supplement 7a*), as well as to removal of all peaks even slightly overlapping TSS or promoter-related annotations (*Figure 3—figure supplement 7b*). It should be noted that chromHMM annotations such as heterochromatin do not imply that a region is constitutively heterochromatic, but instead reflect the most common chromatin state across a large compendium of cell types (*Vu and Ernst, 2022*). Indeed, the fact that we are focusing on ATAC-seq peaks indicates that the chromatin in these regions is accessible in at least one cell type in our study. This result suggests that CREs that are heterochromatic in some cell types may be more prone to large changes in accessibility during evolution (*Figure 3d*), with significant impacts on the *cis*-regulation of nearby genes (*Figure 3e*). Throughout, we refer to CREs assigned to the chromHMM annotation heterochromatin as 'heterochromatin CREs'.

To further investigate the intriguing relationship between heterochromatin ASCA and ASE, we asked whether TSS proximity is an important factor. First, we removed all CREs annotated as promoters and recomputed correlations separately for CREs within 30 kilobases of a TSS (proximal CREs) and those greater than 30 kb away (distal CREs). Interestingly, differences in accessibility in proximal heterochromatin CREs have weak correlations with ASE compared to proximal enhancers (*Figure 3—figure supplement 8a*). However, differences in accessibility in distal heterochromatin CREs were roughly as strongly correlated with ASE as differences in accessibility in enhancer regions (*Figure 3—figure supplement 8b*).

Next, we partitioned genes into cell type-specifically and ubiquitously expressed and recomputed the correlations. While the results for ubiquitously expressed genes mirrored our initial finding of a relatively strong relationship between accessibility in heterochromatin CREs and gene expression, differences in accessibility of heterochromatin CREs were less correlated with ASE for cell-type-specific genes than differences in accessibility of enhancers (*Figure 3—figure supplement 9*). Altogether, our analysis suggests that large changes in accessibility of distal heterochromatin CREs may be particularly important in the *cis*-regulatory evolution of more ubiquitously expressed genes.

We then investigated whether the analog of the relationship between cell-type-specific gene expression and ASE (*Figure 2d*) holds for chromatin accessibility. Since the number of called peaks is largely dependent on sequencing depth (*Figure 3—figure supplement 1*, *Figure 3—figure supplement 10a*), we performed down-sampling to equalize power to detect peaks across cell types (*Figure 3—figure supplement 10b*, Methods). We then called peaks on the down-sampled data and identified peaks as cell-type-specific if they were called as peaks in only one cell type. In agreement with the gene expression data, we observed that cell-type-specific peaks are enriched for ASCA across all cell types and this enrichment generally holds when using varying $\log_2$ fold-change or p-value cutoffs (*Figure 3—figure supplements 11–12*). Analogous to our analysis of gene expression, we also applied a broader definition of cell type-specificity to the ATAC data, in which a peak was considered specific to a cell type if that peak had an absolute $\log_2$ fold-change greater than a chosen threshold (e.g. 0.5 or 1) across all pairwise comparisons with other cell types. We observe strong enrichment for ASCA in cell-type-specific peaks using this definition except when using the most stringent cutoffs due to the very low number of peaks meeting these criteria (*Figure 3—figure supplements 13–14*, Methods). Notably, we observe the same enrichments when controlling for a recently published metric for constraint on non-coding elements that compares the observed vs. expected number of human polymorphisms (*Chen et al., 2022a*), suggesting that differences in evolutionary constraint may not be solely responsible for the observed trends (*Figure 3—figure supplement 15*).

Finally, it is possible that CREs and genes that are less conserved will have more SNPs, and therefore more power to call ASCA and ASE, leading to systematically biased estimates. There is a weak positive correlation between the number of SNPs and the $-\log_{10}(\text{FDR})$ for ASE and a weak negative

or no correlation for ASCA (*Figure 3—figure supplement 16*). Similarly, we observe a weak relationship between the number of SNPs in CREs or genes and absolute log fold-change estimates (*Figure 3—figure supplement 17*). Although the relationship between the number of SNPs and ASE/ASCA is weak, we confirmed that cell-type-specific genes and peaks are still strongly enriched for ASE and ASCA when stratifying by number of SNPs (*Figure 3—figure supplements 18–19*). Overall, our analysis suggests that the result that more cell-type-specific genes and CREs are more evolutionarily diverged is robust to a variety of possible confounders.

We next explored the relationship between cell-type-specific ASCA and ASE. To do this, we developed a novel metric called differential expression enrichment (dEE) to quantify how specific the log fold-change is to a particular cell type or tissue. Our method is based on expression enrichment (EE) (*Yu et al., 2006*), a metric that measures how specific gene expression is to a certain cell type/tissue. dEE estimates how cell-type-specific ASE is for a gene (*Figure 3—figure supplement 20*, Methods) and, analogously, dCAE (differential chromatin accessibility enrichment) measures how cell-type-specific ASCA is for a *cis*-regulatory element (*Figure 3—figure supplement 21*, Methods). dEE and dCAE are high in a cell type if there is a high absolute log fold-change in that cell type and much lower absolute log-fold changes or log fold-changes in the opposite direction in the other cell types. For example, dEE would be close to one for a gene in a cell type if the gene had strongly human-biased ASE in that cell type and very weakly human-biased or chimpanzee-biased ASE in the other cell types. On the other hand, if a gene did not have any strong allelic bias, that gene would have dEE close to zero. dEE is conceptually related to a metric we have previously introduced, diffASE (*Combs et al., 2018*; *Hu et al., 2022*; *York et al., 2018*), and generalizes diffASE to an arbitrary number of cell types and any assay that produces log fold-changes. Using these metrics, we identified 154 instances in which a gene with cell-type-specific ASE (i.e. high dEE) had a peak with cell-type-specific ASCA in the same cell type (i.e. high dCAE, see *Supplementary file 2* for the full list). Of these, 95 showed ASCA and ASE in the same direction which is more than expected by chance (77 expected; p<0.005, binomial test). These results suggest that tissue-specific *cis*-regulatory divergence in chromatin accessibility may often impact tissue-specific gene expression, although this divergence is neither necessary nor sufficient to do so.

## Identifying candidate causal *cis*-regulatory variants by integrating ASE and ASCA across cell types

As high dEE and dCAE in a given cell type might be indicative of a causal link between the change in chromatin accessibility and the change in expression, we focused on the 95 peak-gene pairs with matching direction and used two different strategies to identify examples to investigate in detail. First, we prioritized genes known to play important roles in development. For example, we found that the promoter of *FABP7* has human-biased ASCA specifically in motor neurons (*Figure 4a–b*) and that the *FABP7* gene has human-biased ASE in motor neurons (*Figure 4c*). *FABP7* is used as a marker of glial cells and neural progenitor cells (NPCs) and plays a key role in NPC proliferation and astrocyte function (*Arai et al., 2005*; *De Rosa et al., 2012*; *Ebrahimi et al., 2016*; *Watanabe et al., 2007*). Using previously published single-nucleus RNA-seq data from humans, chimpanzees, and rhesus macaques, we confirmed that *FABP7* shows a human-derived up-regulation in several neuronal subtypes but not glial cells (*Figure 4—figure supplement 1*; *Ma et al., 2022*). To investigate the genetic basis of this cell-type-specific divergence, we leveraged a machine learning model, Sei (*Chen et al., 2022b*), to nominate potentially causal variants in the promoter of *FABP7* (see Methods). Sei is a deep neural network that takes DNA sequence as input and predicts the probability that the sequence has a particular epigenetic state in a variety of cell types and tissues (*Figure 4d*; *Chen et al., 2022b*). While single-base substitutions differing between human and chimpanzee had only minor impacts on predicted *cis*-regulatory activity, 'chimpanizing' the human sequence of the *FABP7* promoter at two small indels (by deleting one base at chr6: 122,779,291 and inserting three bases at chr6: 122,779,115) was sufficient to make the Sei predictions for the chimpanized human sequence closely match the predictions for the complete chimpanzee sequence (*Figure 4e–g*). Making only one of these changes had substantial but weaker effects in both cases, suggesting that both mutations might be functionally important (*Figure 4f and g*). The 1-base insertion in the human lineage introduces a binding site for the neuronally expressed transcription factors GLIS2/3, suggesting a potential molecular mechanism (*Calderari et al., 2018*; *Castro-Mondragon et al., 2022*; *Ke et al., 2015*).

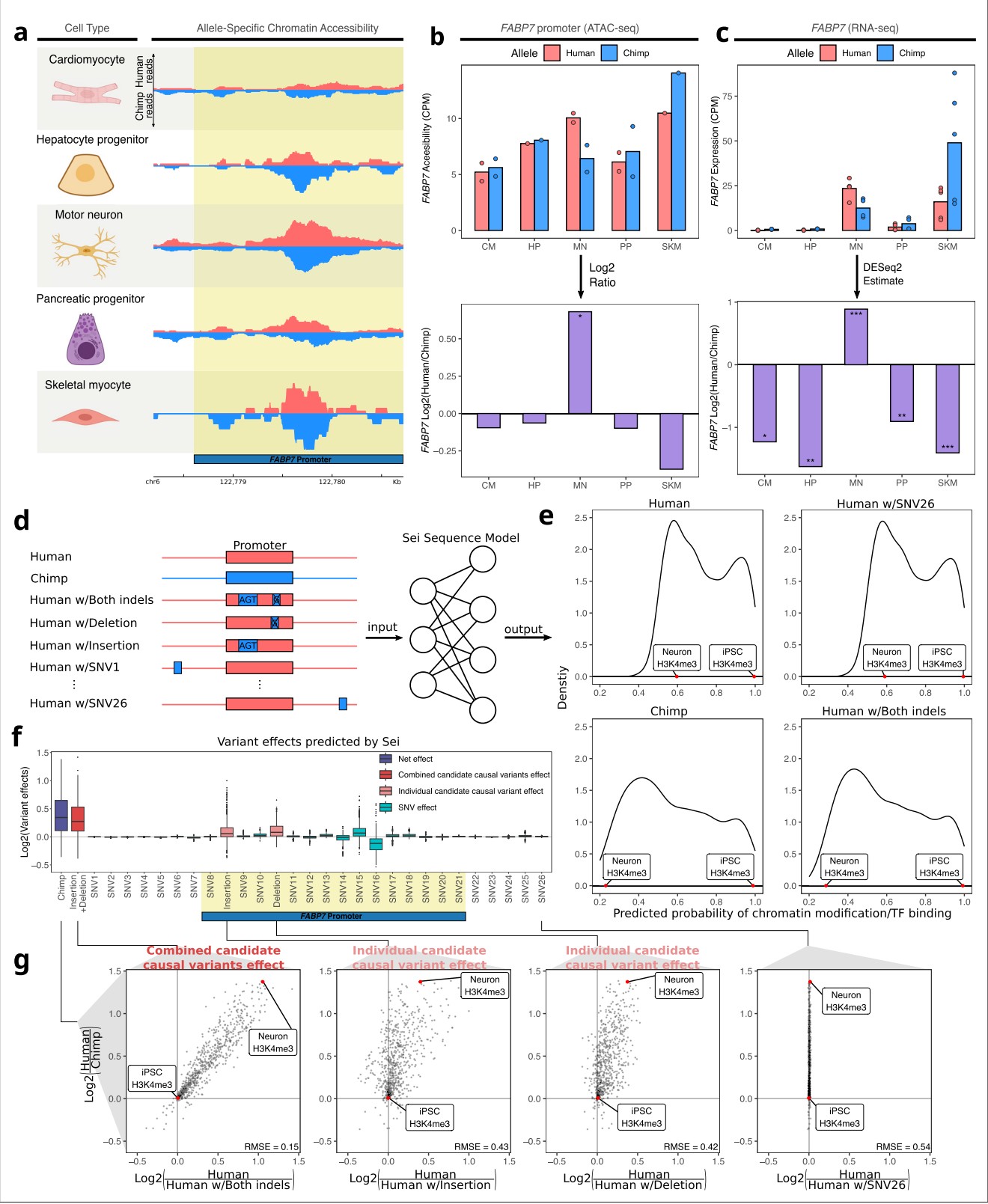

**Figure 4.** Motor neuron-specific human-biased ASE and ASCA for *FABP7* and the promoter of *FABP7*. (**a**) Allelic ATAC-seq tracks are shown in the peak containing the annotated *FABP7* promoter (highlighted in yellow). (**b**) The top panel shows allelic CPM of the *FABP7* promoter across cell types and the bottom panel shows the log fold-change across cell types. (**c**) The top panel shows allelic CPM of the *FABP7* gene across cell types and the bottom panel shows the log fold-change across cell types. (**d**) Outline of the process for variant effect prediction with Sei for *FABP7*. All sequences input to

*Figure 4 continued on next page*

*Figure 4 continued*

Sei were centered at the *FABP7* promoter. The human sequence, chimpanzee sequence, and partially "chimpanized" human sequences (modified by systematically switching the human allele to the chimpanzee allele separately for each human-chimpanzee difference) were fed into Sei to predict the effects of these variants on chromatin state. (**e**) Histogram of the probabilities of various chromatin states and transcription factor binding predicted by Sei was plotted for the human sequence, the chimpanzee sequence, and the human sequence with one human-chimpanzee difference swapped to match the chimpanzee sequence. The human sequence with SNV26 changed to the chimpanzee allele and the human sequence switched at both indels are shown as examples. A histone modification (H3K4me3) predicted in two cell types was labeled to illustrate how the predictions depend on both input sequence and cell type. (**f**) Plot of the predicted effects of all single nucleotide differences and indels between the human and chimpanzee genomes in the Sei input window (the *FABP7* promoter is highlighted in yellow). (**g**) Scatterplots showing the correlation of the effects of both indels (left panel), each individual indel (middle two panels), and a representative SNV (right panel) on Sei predictions with the difference in Sei predictions between the human and chimpanzee sequences. The root mean square error (RMSE) was computed and shown in each figure.

The online version of this article includes the following figure supplement(s) for figure 4:

**Figure supplement 1.** Differential *FABP7* expression across cortical cell types.

As another approach to ranking the 95 peaks, we searched for peaks containing human-chimpanzee sequence differences in otherwise highly conserved genomic positions, since these could reflect changes in selective pressure. Using PhyloP scores for 241 placental mammals (*Sullivan et al., 2023*) to assess conservation, one of the top-ranked peaks was a putative enhancer six kilobases away from the TSS of *GAD1*, which plays a key role in the synthesis of the neurotransmitter GABA (*Feldblum et al., 1993*). Notably, part of this peak has been classified as a human accelerated region (HAR; *Girskis et al., 2021*; *Hubisz and Pollard, 2014*; *Pollard et al., 2006*)—a short sequence that is highly conserved in mammals yet contains an unusual number of human-specific mutations. Both the accessibility in the peak and *GAD1* expression are only chimpanzee-biased in motor neurons (*Figure 5a–c*, *Figure 5—figure supplement 1*). Applying Sei to estimate the predicted effect of every variant in this region, we found that a single-nucleotide substitution within the HAR (chr2: 170,823,193) has by far the largest predicted *cis*-regulatory effect and most closely matches the differences in Sei predictions between the full human and chimpanzee haplotypes (*Figure 5d–g*). Interestingly, this mutation is predicted to disrupt a binding site for several basic helix-loop-helix transcription factors that play essential roles in neuronal differentiation such as Ascl1 (*Figure 5—figure supplement 2*; *Castro-Mondragon et al., 2022*; *Mizuguchi et al., 2006*; *Yang et al., 2017*).

As *GAD1* is only highly expressed in GABAergic neurons (and was therefore lowly expressed in the cell types studied here, *Figure 5—figure supplement 3a*), we investigated whether this reduced expression of human *GAD1* also occurs in cortical organoids which contain GABAergic neurons together with other cell types in which *GAD1* is not highly expressed. We analyzed our previously published data from human-chimpanzee hybrid cortical organoids (*Agoglia et al., 2021*) and found that the expression of *GAD1* from the chimpanzee allele spikes higher than that of the human allele around day 50 of hybrid cortical organoid differentiation before dropping in expression over time to match the human expression level (*Figure 5—figure supplement 3b*). Because ASE in the hybrid cells controls for any potential interspecies differences in differentiation kinetics or cell type composition, this difference must be the result of *cis*-regulatory divergence between humans and chimpanzees. This expression difference is also more pronounced in comparisons of human and chimpanzee parental cortical organoids, with a higher absolute log fold-change at day 50, day 100, and day 150, only returning to equal expression at day 200 (*Figure 5—figure supplement 3c*). While this could be due to differences in cell type proportion between human and chimpanzee organoids, it might also be due to a reinforcing *trans*-acting effect.

To test whether this difference in expression also occurs specifically during GABAergic neuron differentiation we examined *GAD1* expression in single cell RNA-seq data from human and chimpanzee cortical organoids (*Kanton et al., 2019*). Consistent with our cortical organoid results, we observed a peak in *GAD1* expression in less mature chimpanzee GABAergic neurons that is absent in the corresponding part of the trajectory in human neurons (*Figure 5—figure supplement 4*). Notably, a similar trend holds regardless of which GABAergic sub-trajectory (i.e. equivalent to GABAergic neurons from the caudal, lateral, or medial ganglionic eminences) is examined suggesting this difference is not unique to a particular type of GABAergic neuron (*Figure 5—figure supplement 4*). Finally, we examined the accessibility of the putative *GAD1* enhancer more closely. Consistent with a potential role for this enhancer in the spike in *GAD1* expression during development, the accessibility of this

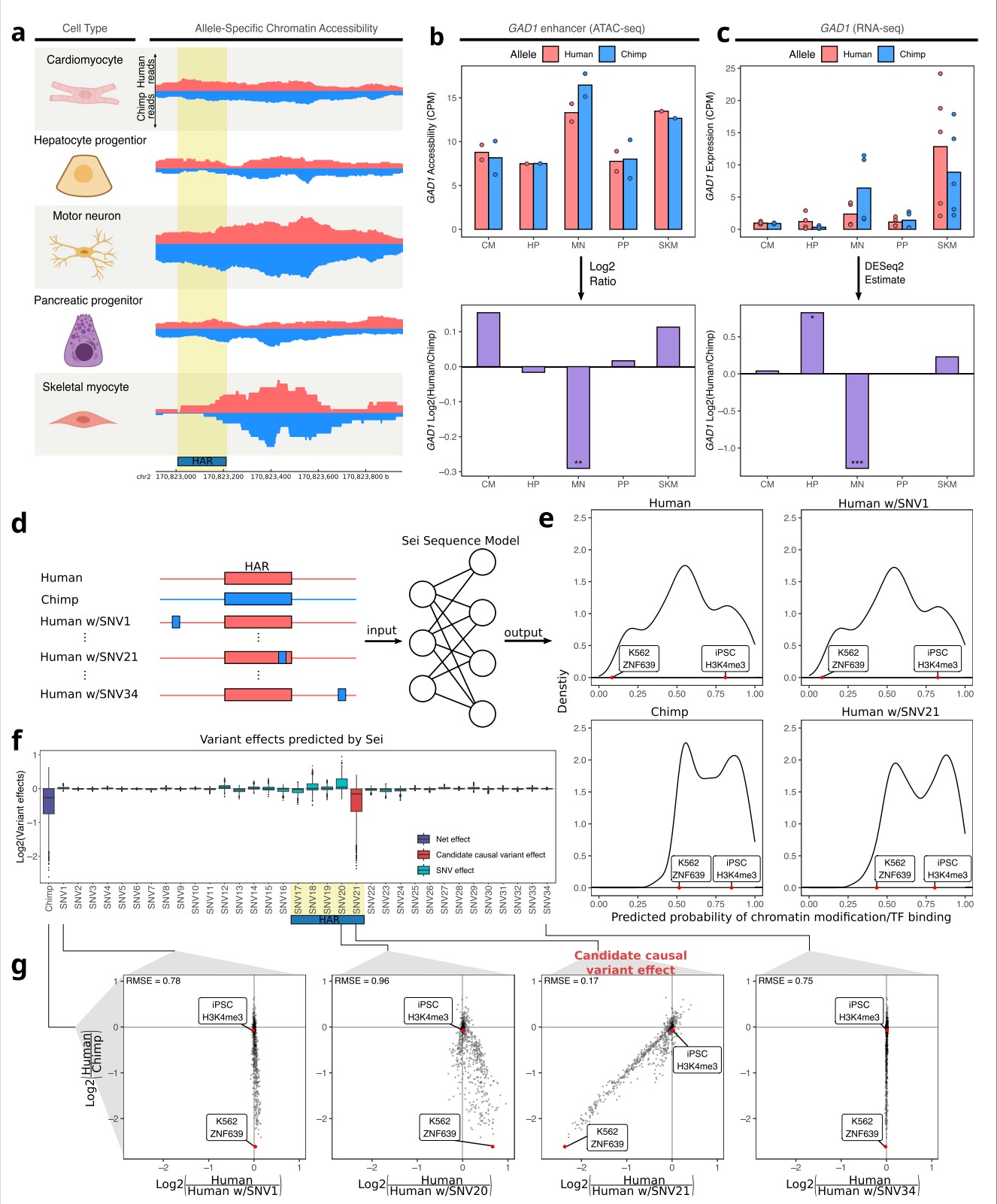

**Figure 5.** Motor neuron-specific chimpanzee-biased ASE for *GAD1* and ASCA for a HAR near the *GAD1* TSS. (**a**) Allelic ATAC-seq tracks are shown for the peak near the *GAD1* TSS that contains a HAR (highlighted in yellow). (**b**) The top panel shows allelic CPM of the CRE near the *GAD1* TSS across cell types and the bottom panel shows the log fold-change across cell types. (**c**) The top panel shows allelic CPM of the *GAD1* gene across cell types and the bottom panel shows the log fold-change across cell types. (**d**) Outline of the process for variant effect prediction with Sei for *GAD1*. All input

*Figure 5 continued on next page*

*Figure 5 continued*

sequences to Sei were centered at the HAR. The human sequence, the chimpanzee sequence, and modified sequences with the human sequence altered at each substitution to match the chimpanzee sequence were fed into the Sei sequence model to predict the effects of these variants on the chromatin state. (**e**) Histogram of the probabilities of various chromatin states and transcription factor binding predicted by Sei was plotted for the human sequence, the chimpanzee sequence, and two examples in which the human sequence with only one SNV "chimpanized" (human w/SNV) was input to Sei. The histogram of the probability of the sequence having a particular epigenomic annotation (predicted by Sei) was plotted for human, chimpanzee, human w/SNV1 changed to match the chimpanzee sequence, and human w/SNV21 changed to match the chimpanzee sequence. Two epigenomic annotations were labeled as examples that show the different values output by Sei with these two different sequence inputs. (**f**) Plot of the predicted effects of all single nucleotide differences between human and chimpanzee in the Sei input window centered at the HAR (highlighted in yellow). Positions were switched to the chimpanzee allele individually. (**g**) Scatterplots showing the correlation of the effects of four SNVs on Sei predictions with the difference in Sei predictions between the human and chimpanzee sequences. The root mean square error (RMSE) was computed and shown in each figure.

The online version of this article includes the following figure supplement(s) for figure 5:

**Figure supplement 1.** A HAR near the TSS of *GAD1* is chimpanzee-biased only in motor neurons.

**Figure supplement 2.** A human-specific SNP disrupts a potential Ascl1 binding site.

**Figure supplement 3.** *GAD1* ASE and differential expression across cell types and in cortical organoids.

**Figure supplement 4.** *GAD1* expression across GABAergic neuron differentiation in human and chimpanzee cortical organoids (*Kanton et al., 2019*).

**Figure supplement 5.** Chromatin accessibility of the HAR near the *GAD1* TSS across human striatal organoid development.

enhancer mirrors the expression of *GAD1* in human cortical and striatal organoids (*Figure 5—figure supplement 5*) with high accessibility between day 50 and day 100 before decreasing somewhat near day 150 (*Trevino et al., 2020*). Overall, our results demonstrate how the combination of RNA-seq, ATAC-seq, and machine learning models can nominate variants that may be responsible for cell-type-specific changes in gene expression and chromatin accessibility.

## Discussion

In this study, we quantified human-chimpanzee *cis*-regulatory divergence in gene expression and chromatin accessibility in six different cell types representing diverse developmental lineages. Across the thousands of genes with ASE, we found that most *cis*-regulatory divergence is specific to one or a few cell types. Furthermore, we found that divergent *cis*-regulation is linked to tissue-specificity, with tissue-specific genes being enriched for ASE and tissue-specific regulatory elements being enriched for ASCA. As this result was largely unchanged when stratifying by evolutionary constraint, our results suggest that changes in the expression of genes with more cell-type-specific expression patterns may be less deleterious than changes in more broadly expressed genes, supporting the idea that cell-type-specific divergence may be less pleiotropic (*Wittkopp and Kalay, 2011*). Overall, this suggests that broad changes in expression in cell type-specifically expressed genes may be an important substrate for evolution, but it remains unclear whether positive selection or lower constraint plays a larger role in driving the faster evolution of more cell type-specifically expressed genes. Future work will be required to more precisely quantify the relative roles of positive selection and evolutionary constraint in driving changes in gene expression.

We also identified several sets of genes evolving under lineage-specific selection that may have played a role in establishing unique facets of human physiology and behavior. Most interestingly, we found evidence for selection on sodium ion transporters and syntaxin binding proteins that may alter the electrophysiological properties of motor neurons, and potentially other types of neurons as well (*Brose et al., 2019*; *Meisler et al., 2021*). The complexity of the molecular machinery regulating neuronal excitability and synaptic vesicle release make it difficult to say what the effects of these gene expression changes are on the excitability of motor neurons without electrophysiology data from human and great ape neurons coupled with perturbation of candidate genes. However, given the divergence in locomotion and motor skills between humans and chimpanzees, one could speculate that these changes may have had some role in the evolution of motor control and learning in humans.

In this work, we developed two metrics—dEE and dCAE—to quantify the degree of cell-type-specific differential expression and accessibility. These metrics are largely analogous to widely used metrics that quantify tissue- or cell-type-specific expression level and applicable to any comparison of log fold-changes across conditions. They markedly improved our ability to identify matching

cell-type-specific ASE and ASCA and led to the identification of 95 peak-gene pairs that had highly cell-type-specific concordant changes in accessibility and expression.

One such example is a human-derived increase in *FABP7* expression in several types of human neurons. As *FABP7* is not highly expressed in adult mouse neurons, the functional consequences of its higher expression in human neurons are difficult to predict (*Yao et al., 2021*). *FABP7* plays a role in the uptake of the fatty acid Docosahexaenoic Acid (DHA), an important component of neuronal membranes (*Akbar et al., 2005*; *Choi et al., 2021*). DHA promotes neuronal survival through phosphatidylserine accumulation, so it is possible that the human-specific *FABP7* expression increases neuronal DHA uptake leading to reduced apoptosis in human neurons during development and ultimately contributing to a larger number of neurons in humans (*Akbar et al., 2005*; *Choi et al., 2021*).

In addition, we identified a highly conserved developmentally dynamic enhancer near *GAD1* that may have partially lost activity in the human lineage resulting in a decrease in *GAD1* expression early in GABAergic neuron development. By integrating with the deep learning model Sei (*Chen et al., 2022b*), we identified a variant that may account for the chimpanzee-biased ASCA in this region. Interestingly, the ASE of *GAD1* was coupled with a relatively small (though significant) magnitude of ASCA. This could potentially reflect divergence in transcription factor binding that leaves a 'footprint' resulting in subtle ASCA (*Vierstra et al., 2020*). Overall, our data suggest that this enhancer has lost activity in the human lineage, potentially altering the expression pattern of *GAD1* during neuronal development. *GAD1* is the rate-limiting enzyme for GABA synthesis so GABA levels are likely responsive to changes in *GAD1* expression (*Feldblum et al., 1993*). GABA release has complex context-specific effects on neurodevelopment, making it difficult to speculate as to what the phenotypic effects of reduced GABA synthesis during human neurodevelopment might be (*Ben-Ari et al., 2012*). However, the high conservation of this *cis*-regulatory element in placental mammals implies that its human-specific disruption is likely to have important neurodevelopmental effects. Careful perturbation of this enhancer and *GAD1* expression in mouse models will be required to explore this further.

In addition to following up on our findings on *GAD1* and *FABP7*, there are other exciting future directions for this work. First, additional bulk assays such as those that measure methylation, chromatin conformation, and translation rate could lead to a better understanding of what molecular features ultimately lead to cell-type-specific changes in gene expression. Furthermore, the use of deep single-cell profiling of hybrid lines derived from iPSCs from multiple individuals of each species during differentiation could enable the identification of many more highly context-specific changes in gene expression and chromatin accessibility such as the differences in *GAD1* we highlighted here. Finally, integration with data from massively parallel reporter assays and deep learning models will help us link specific variants to the molecular differences we identified in this study.

Overall, our study provides foundational data, insight, and computational tools that will improve our understanding of cell-type-specific *cis*-regulatory evolution and the role it has played in the establishment of human-specific traits.

## Methods
### Generation of multiple human-chimpanzee hybrid cell types

We used two previously described human-chimpanzee hybrid iPS cell lines (hybrid1 and hybrid2, previously denoted Hy1-25 and Hy1-30 respectively) (*Agoglia et al., 2021*). These lines were confirmed to be free of mycoplasma contamination and their identity was confirmed using the RNA sequencing data generated in this study. Before differentiation, cells were routinely cultured on matrigel in mTeSR1 or mTeSR Plus (Stem Cell Technologies cat #85850 or cat #100–0276). Culture and *in vitro* differentiation of iPS cells into six cell types (motor neurons [MN], cardiomyocytes [CM], hepatocyte progenitors [HP], pancreatic progenitors [PP], skeletal myocytes [SKM], and retinal pigment epithelium [RPE]) was carried out by the Columbia Stem Cell Core Facility using published protocols (*Burridge et al., 2014*; *Chal et al., 2016*; *Korytnikov and Nostro, 2016*; *Mallanna and Duncan, 2013*; *Maury et al., 2015*; *Sharma et al., 2019*).

### Preparation of RNA-seq libraries

All samples were cryopreserved in liquid nitrogen before RNA extraction (*Milani et al., 2016*). Cells were gently thawed and then washed with PBS and cell pellets were collected via centrifugation

at 1000 RPM for 5 min. Cell pellets were loosened by flicking the tube and an appropriate volume of Buffer RLT based on the cell count were added following the RNeasy Mini Kit (Qiagen, 74104) protocol. Total RNA extraction and on-column DNase digestion were performed using RNeasy Mini Kit (Qiagen, 74104) and RNase-Free DNase Set (Qiagen, 79254). RNA quality was assessed using the Agilent Bioanalyzer RNA Pico assay. Only samples with an RNA integrity number (RIN) greater than or equal to 7 were used to prepare cDNA libraries. All RNA-seq libraries except three motor neuron libraries were prepared using the TruSeq Stranded mRNA kit (Illumina, 20020594) and the TruSeq RNA CD Index Plate (Illumina, 20019792) from between 100 ng and 1 µg total RNA following the manufacturer's protocols. Due to low yield of total RNA, three motor neuron libraries (2 hybrid2 and 1 hybrid1) were prepared using Illumina Stranded mRNA Prep (Illumina, 20040532) and IDT for Illumina RNA UD Indexes Set A, Ligation (Illumina, 20040553). Notably, the four motor neuron libraries did not cluster by library preparation method. All libraries were normalized, pooled at an equimolar ratio using Qubit measurements, and sequenced on an Illumina HiSeq 4000 to generate 2x150 bp paired-end reads.

## Identification of confident human-chimpanzee SNVs

To identify a confident list of human-chimpanzee SNVs that could be used to quantify allele-specific expression and chromatin accessibility, we first downloaded hg38-panTro6 MAF files from UCSC and whole-genome sequencing data generated from the parental human and chimpanzee iPS cells (in the form of bam files aligned to hg38 and panTro5, generously provided by the Gilad lab). We first converted them back to fastq files and then mapped reads to panTro6 and hg38 (we mapped both human and chimpanzee to both reference genomes) using bowtie2 with the flags —very-sensitive-local -p 16 (*Langmead and Salzberg, 2012*). We then used a modified version of our previous approach to filter out SNVs that could not be confidently identified as homozygous in the human and chimpanzee parental lines (*Agoglia et al., 2021*). Briefly, we extracted SNVs and indels from both human and chimpanzee MAF files, counted reads in the WGS data that supported the human, chimpanzee, or an alternative base at that position, then filtered out any SNVs with <2 reads or <90% of reads supporting that species' base. We then reformatted files, merged with indels for use in Hornet, and generated a modified bed file of SNVs that includes the human and chimpanzee base at the SNV position (*van de Geijn et al., 2015*).

## Generation of allele-specific count tables

An allele-specific expression pipeline adapted from Agoglia et al. and our updated high-confidence SNV list was used. The whole pipeline was carried out twice independently using hg38 or panTro6 as the reference genome. This approach was taken to eliminate genes showing strong mapping bias, defined here as genes with an absolute difference in $\log_2$ fold-change between the panTro6-referenced and hg38-referenced runs greater than one. All sequencing reads were trimmed with SeqPrep (adapters specified by the manufacturer for the different library preparation kits) and mapped using STAR with two passes and the following parameters: `--outFilterMultimapNmax 1` (*Dobin et al., 2013*; *John St. John, 2024*). Uniquely aligned reads were deduplicated with Picard and Hornet (an implementation of WASP which first removes reads overlapping indels) was used to correct for mapping bias (*Broad Institute, 2024*; *van de Geijn et al., 2015*). Reads were assigned to either the human allele if they contained one or more human-chimpanzee single nucleotide differences that matched the human sequence and zero positions that matched the chimpanzee sequence (and vice versa for assigning reads to the chimpanzee allele) and counted per gene as previously described (*Agoglia et al., 2021*).

## Detection of aneuploidy on chromosome 20 and slight chimpanzee parental contamination in PP hybrid2 samples

In our quality control process, we plotted the log2 fold-change for each gene along every chromosome and inspected the results. This revealed a clear bias toward the human allele on a part of chromosome 20 for hybrid2 samples, suggesting chromosome 20 aneuploidy which was also reported by *Agoglia et al., 2021*. As a result, we excluded chromosome 20 from all downstream analyses. In addition, we found that PP hybrid2 samples had a slight bias toward the chimpanzee allele across every chromosome which was most likely due to a small fraction of contaminating chimpanzee cells

in these samples. Rather than removing these samples, we normalized the allele-specific count tables by subtracting a small number of reads from the chimpanzee allele counts calculated based on the biased ratio summarized from genome wide human and chimpanzee allele counts, to force a global log fold-change (across all autosomes except chromosome 20) of zero between the human and chimpanzee alleles. We applied this normalization to all other samples as well. To evaluate the success of this strategy, we simulated the effects of chimpanzee iPSC contamination and our subsequent correction. Specifically, using hybrid and chimpanzee parental iPSC RNA-seq from *Agoglia et al., 2021* from the same iPS cell lines as used in this study, we simulated chimpanzee iPSC contamination by mixing chimpanzee RNA-seq data into hybrid2 data to reach a similar degree of chimpanzee bias level to that observed in the PP hybrid2 samples. We then identified genes showing ASE (see 'Identifying genes with ASE') using the counts from the original hybrid samples, simulated contaminated samples, and corrected simulated contaminated samples and compared the outputs (*Figure 1—figure supplement 4*).

## PCA and hierarchical clustering

Allelic counts were normalized by DESeq2 rlog and principal components analysis (PCA) was performed on rlog normalized allelic counts with default centering and scaling (*Love et al., 2014*). The top 1000 variable genes with the highest variance of normalized allelic counts across all cell types were used to compute Euclidean distance matrices. The R package pheatmap was used to do hierarchical clustering and heatmap plotting.

## Identifying genes with ASE

DESeq2 was used to measure allele-specific expression (ASE) in each cell type (*Love et al., 2014*). All reads from chromosome 20 were removed (as mentioned above). Two replicates per hybrid line per cell type (plus one additional replicate for SKM hybrid2 for a total of three samples) were used by DEseq2 with model ~hybLine + Species to measure differential expression level. A likelihood ratio test (test="LRT", betaPrior = FALSE) was used to compute p-values. p-Values were then false discovery rate adjusted using an implementation of the Benjamini-Hochberg correction in the R package qvalue (*Benjamini and Hochberg, 1995*; *Storey Lab, 2024*). Log fold-changes were shrunk as recommended by the DESeq2 pipeline (*Zhu et al., 2019*). Differentially expressed genes were defined as those with FDR <0.05 when aligned to hg38 and panTro6 as well as an absolute difference in log fold-change ≤ 1 when comparing the results from the two alignments.

## Identifying cell type-specifically expressed genes

For the more traditional definition of cell-type-specific genes, we required transcripts per million (TPM) <1 for a gene in every cell type except one. In the cell type with TPM >1, we varied how highly expressed the gene had to be in that cell type (again using a TPM cutoff, varying between one and five) to consider that gene to be specific to that cell type. A similar process to the one described in 'Identifying differentially expressed genes' was used to identify cell-type-specific genes based on the broader definition described in the main text. Rather than using allelic counts, total counts for each sample (i.e. all uniquely mapping deduplicated reads regardless of their allelic origin) were computed by summing all allelic and non-allelic counts. These counts were inputted to DESeq2 and the expression of each gene was compared pairwise between all cell types (*Love et al., 2014*). Genes were defined as cell type-specifically expressed in a cell type only if all pairwise comparisons between that cell type and other cell types resulted in an FDR <0.05 using both hg38 and panTro6 aligned counts. Due to the markedly lower number of differentially expressed genes identified in SKM, results were computed both including and excluding SKM. An analogous procedure was used to identify more broadly defined cell-type-specific peaks in the down-sampled ATAC-seq dataset. Peaks were defined as specific to a cell type if the absolute log fold-change was greater than 0.5 across all pairwise comparisons with the other cell types. We also tested an absolute log fold-change threshold of 1 to ensure that our results were not sensitive to the choice of cutoff.

## Enrichment test for genes with cell-type-specific expression patterns and genes showing ASE

Odds ratios were calculated using the unconditional maximum likelihood estimate implemented in the R package epitools function oddsratio.wald(), and 95% confidence intervals and p-values were calculated using the normal approximation. A directly analogous procedure was performed to test for enrichment of peaks with ASCA and cell-type-specific peaks.

## Enrichment test stratified by expression level or evolutionary constraint

Enrichment tests were carried out as in 'Enrichment test of cell-type-specific expression patterns and genes showing ASE' except that genes were split into five equal size bins depending on which factors were used to stratify genes, and tests were done in each bin. When stratifying by expression level, genes were ordered in ascending order based on expression level (TPM) and then split into five equally sized bins where genes in the 0–20% bin are the most lowly expressed genes and genes in the 80–100% bin are the most highly expressed genes. When stratifying by constraint metrics such as ASE variance (see 'Identification of lineage-specific selection on gene expression' for details on how ASE variance was computed) or pHI, genes were ordered in ascending order based on ASE variance values and then split into five equal size bins where the 0–20% bin contains genes with the lowest ASE variance (i.e. most evolutionarily constrained) and the 80–100% bin contains genes with highest ASE variance (i.e. least evolutionarily constrained). To stratify the ATAC data by constraint, we used the 'QCed genomic constraint by 1 kb regions' computed by the gnomAD consortium (*Chen et al., 2022a*). The gnomAD consortium computed the ratio of the observed and expected number of human polymorphisms in 1 kb regions tiling the human genome and then converted this metric to a z-score (see *Chen et al., 2022a* for additional details). We further removed any regions that overlapped protein-coding exons from the human gtf file using bedtools subtract (*Quinlan and Hall, 2010*). If a peak overlapped two or more 1 kilobase windows, it was assigned to the window with the highest constraint, mirroring the procedure used by the gnomAD consortium to assign peaks to ENCODE regulatory elements (*Chen et al., 2022a*). Once the peaks were ranked by this metric, a procedure identical to that for the gene expression constraint metrics outlined above was performed.

## Identification of lineage-specific selection on gene expression

We developed a modified version of our previously published pipeline, which uses ASE values from many individuals of a single species to estimate *cis*-regulatory constraint of each gene (*Starr et al., 2023*). We restricted these ASE values to GTEx samples from the tissue(s) of origin for each cell type (hepatocytes with liver, skeletal muscle with skeletal muscle, cardiomyocytes with the left ventricle of the heart, and pancreatic progenitors with the pancreas). As RPE and MN did not have clear matching tissues (e.g. GTEx does not include data from the eye), we compared RPE to all GTEx samples and MN to all brain and peripheral nerve samples. We then used the Mann-Whitney U test to compare the human population ASE distribution to the human-chimpanzee ASE distribution as previously described (*Starr et al., 2023*). We employed the previously described signed ranking by Mann-Whitney p-value that incorporates whether a gene has human or chimpanzee-biased ASE with GSEAPY and the binomial test to identify instances of lineage-specific selection (*Starr et al., 2023*). Positive selection on a gene set is only inferred if there is statistically significant human- or chimpanzee-biased ASE in that gene set (using an FDR-corrected p-value from the binomial test). Due to the focus on tissue-specificity, we did not filter redundant gene sets with GSEAPY FDR <0.25 in multiple cell types (*Starr et al., 2023*; *Subramanian et al., 2005*). To compute ASE variance, we used the same tissue-of-origin-matched data from GTEx and computed the variance of the ASE ratios after filtering out samples with fewer than ten counts from the reference or alternate allele (see *Starr et al., 2023* for additional details). For example, if a sample had 11 reference counts and 2 alternate counts for a gene, that sample would be excluded for that gene.

## Preparation of ATAC-seq libraries

We used the OmniATAC protocol with the only modification being the use of 25,000 cells instead of 50,000 since the fused iPS cells are tetraploid (*Corces et al., 2017*). All samples for ATAC-seq prep were from the same vials used in RNA-seq library preparation except for the motor neuron libraries

due to the low yield of total RNA extracted from motor neurons. After library preparation and running samples on a Bioanalyzer, we noticed a considerable number of fragments greater than 1000 bases in length. To reduce these fragments, we size selected with Ampure beads using the protocol from the Kaestner lab available here. After size selection and rerunning on the Bioanalzyer, we pooled the libraries together and sequenced them to compute quality control metrics. We used the R package ChrAccR to compute TSS enrichment scores (*Mueller, 2024*). We pooled all libraries with TSS enrichment score greater than 3.5. This resulted in 2 CM libraries, 2 MN libraries, 2 PP libraries, 1 SKM library, and 1 HP library. After pooling, libraries were sequenced on an Illumina Hiseq 4000 to produce 2x150 paired-end reads.

## Mapping the ATAC-seq data

We trimmed reads using SeqPrep and then mapped them to the hg38 and panTro6 reference genomes with bowtie2 in paired-end mode (*John St. John, 2024*; *Langmead and Salzberg, 2012*). The following parameters were used: `-X 2000 --very-sensitive-local -p 16`. After mapping, duplicates were removed via Picard MarkDuplicates (*Broad Institute, 2024*). We then removed multi-mapping reads with the command samtools `view -b -q 10` (*Li et al., 2009*). Due to the format of bowtie2's output, running Hornet on all reads at once was excessively RAM intensive. Therefore, we split the bam files by chromosome and ran Hornet on each of the chromosomes separately. We used the files of SNVs and indels generated as described above as input to Hornet. After Hornet finished running, we used samtools merge to merge all autosomes and sex chromosomes (we excluded the mitochondrial genome) to create a final bam file for downstream analysis (*Li et al., 2009*).

## Peak calling and filtering

As only one replicate was available for SKM and HP, we generated two pseudo-replicates by randomly assigning reads to one of two files using Picard SplitSamByNumberOfReads (*Broad Institute, 2024*). We then called peaks on each file separately, as well as a merged file containing all the reads from a particular cell type. For example, for MN, both replicates were pooled and peaks were called on that file as well as the two replicates separately. Before peak calling, all bam files were converted to bed files. We called peaks using MACS2 callpeak with the following arguments: `-f BED -p 0.01 --nomodel -shift 75 --extsize 150 -B --SPMR --keep-dup all --call-summits` (*Zhang et al., 2008*, p. 2). We called peaks on both the chimpanzee-referenced and human-referenced bam files. After peak calling, we sought to filter peaks using a modified version of the ENCODE pipeline designed to eliminate peaks that lack a one-to-one ortholog between humans and chimpanzees. The following pipeline was run on each cell type separately. We first filtered peaks that were not called in both replicates as well as the pooled file using code from the ENCODE pipeline based on bedtools and awk (*Quinlan and Hall, 2010*). We then used a custom Python script to merge overlapping peaks and used UCSC LiftOver to lift the peaks from hg38 to the panTro6 and back to hg38 as well as from panTro6 to hg38 (*Kuhn et al., 2013*). We then used bedtools to intersect the resulting human referenced files and filtered out any peaks that did not have at least 25% overlap with a peak in the other file (*Quinlan and Hall, 2010*). After filtering out peaks overlapping ENCODE blacklisted regions and merging overlapping peaks again, we lifted the file that was originally chimpanzee-referenced back to the chimpanzee genome (*Amemiya et al., 2019*). Finally, we removed human-referenced peaks if their chimpanzee-referenced counterpart failed to lift over (*Kuhn et al., 2013*). As only a relatively small number of peaks failed to lift over (e.g. because the region was split in the new genome), any peaks that failed to lift over were excluded.

## Annotating the peak lists

To annotate the peaks, we used the list of TSSs defined by Horlbeck et al. to annotate peaks (*Horlbeck et al., 2016*). We lifted over each TSS to hg38, expanded 1000 bases on either side of the midpoint of each TSS to generate promoters, and merged any promoters that overlapped while retaining all unique gene names associated with the promoter (*Kuhn et al., 2013*). We then used reciprocal LiftOver with panTro6 to filter out non-orthologous promoters and used bedtools intersect to link peaks to promoters and expanded the peak to include the entirety of the promoter if necessary (*Quinlan and Hall, 2010*). Through this process, we also compiled a list of non-promoter CREs (sometimes labeled as enhancers as enhancers are thought to be the most common type of CRE). We used

bedtools closest to link these non-promoter CREs to the two closest protein coding genes (*Quinlan and Hall, 2010*). Notably, the gene naming conventions differ for the Horlbeck et al. TSS list and the GTF file used for RNA-seq processing. We altered all gene names in peaks to match those found in the GTF file. In some cases, the gene no longer existed in the updated hg38 GTF in which case the gene name was replaced with NAN.

## Generating a unified peak list

We next merged our cell-type-specific peak list across all five cell types to create a unified peak list. To do this, we iteratively intersected all the peaks with bedtools and then merged any overlapping peaks (*Quinlan and Hall, 2010*). Finally, we added back any peaks that did not intersect a peak found in any other cell types. We then took the chimpanzee and human-referenced versions of these peak lists and ran them through the LiftOver-based non-homologous peak filtering pipeline described above to generate a final file of all identified peaks as well as which cell type (s) they were called in *Kuhn et al., 2013*. Then, we reran the annotation pipeline described in 'Annotating the peak lists' on this new set of peaks. In total, this process resulted in 251,669 ATAC peaks.

## Counting reads in peaks and further peak filtering

First, we split the bam files into reads that we could confidently assign to the chimpanzee genome and reads we could assign to the human genome. We used our bed file of high-confidence SNVs and required at least one SNV matching the human genome as well as no SNVs matching the chimpanzee genome for a read to be assigned to human (and vice versa for chimp). We then used a custom Python script to reformat the peak list bed files as GTF files and used HTSeq to count reads in peaks using the following parameters: -s no -m union -r pos (*Anders et al., 2015*). We only kept peaks if they had a mean read count across replicates within a cell type of at least 25 from either allele. For example, if a peak has an average of 27 reads from the human allele and an average of 10 reads from the chimpanzee allele in MN, that peak would be kept in MN. On the other hand, if the same peak had an average of 24 reads from the human allele and an average of 10 reads from the chimpanzee allele in CM, that peak would be discarded for CM.

We next filtered the reads to remove peaks that might be differentially accessible but show evidence of mapping bias or do not agree between replicates. To do this, we removed any peaks with an absolute $\log_2$ fold-change greater than one in one replicate but with a fold-change of any magnitude in the opposite direction in the other. This was not done for SKM or HP as we had only one replicate. We then removed any peaks that had a log fold-change in opposite directions with an absolute difference greater than 1 in at least one replicate when comparing the human-referenced and chimpanzee-referenced counts. Finally, as described in section "Detection of aneuploidy on chromosome 20 and slight chimpanzee parental contamination in PP hybrid2 samples' for RNA-seq data analysis, we removed any peaks on chr20 and took this as our final list of peaks for downstream analyses. Allelic counts were normalized as described in the RNA-seq data analysis. We tested for allele-specific chromatin accessibility (ASCA) using the binomial test applied to the normalized allelic counts (summed by species within a cell type). We considered any peaks with a binomial p-value less than 0.05 to be nominally differentially accessible. After this filtering, we retained 76,360 peaks for additional analysis. Using the set of peaks that passed filtering for each cell type, we plotted the total number of promoter and non-promoter peaks that passed filtering in each cell type as well as the number of peaks per gene.

## Down-sampling to identify cell-type-specific ATAC-seq peaks

As the number of peaks detected by ATAC-seq is generally a function of read depth and our read depth varied widely across cell types, we restricted to one replicate (always hybrid1 if two replicates were available) and down-sampled reads to match the SKM sample with lowest sequencing depth. We then called peaks for cell types with a single ATAC replicate as described above.

## Allelic chromatin accessibility tracks

Allelic bam files with reads originating from the human allele and the chimpanzee allele (respectively) were obtained as described in 'Counting reads in peaks and further peak filtering'. Two replicates in CM, MN, and PP were pooled by cell type. Bam files were converted into bigWig files using the

python package deepTools bamCoverage with options: `--binSize 1 --normalizeUsing CPM --effectiveGenomeSize 2862010578 --ignoreForNormalization chr20 --extendReads` (*Ramírez et al., 2014*). Tracks were visualized and plotted using the python package pyGenomeTracks (*Lopez-Delisle et al., 2021*). When comparing human and chimpanzee log fold-change track differences in each cell type, deepTools bigwigCompare was used to compare between human bigWig and chimpanzee bigWig with options: `--pseudocount 1 --skipZeroOverZero --operation log2 -bs 1` (*Ramírez et al., 2014*).

## ChromHMM annotation and correlation with ASE

A universal chromHMM annotation was obtained for each peak based on overlap with any of the 15 categories in chromHMM (excluding the blacklist category, for which peaks had already been removed) (*Ernst and Kellis, 2017*; *Vu and Ernst, 2022*). Divergence was measured as the z-score transformed median of the absolute $\log_2$ fold-change of human and chimpanzee normalized counts in each peak. Each peak was assigned to the closest gene and then Pearson correlation was computed between the chromatin accessibility $\log_2$ fold-change and the expression $\log_2$ fold-change for each peak and its nearest gene.

Pearson correlation was computed only on categories including at least 15 peaks. When showing results for differentially accessible peaks, only peaks with binomial p-values less than 0.05 were kept and used in computing the Pearson correlation. When assigning a unique chromHMM to each peak, the chromHMM category that covered the largest portion of each peak was used. When filtering out promoter-related annotations, peaks covering any promoter-related chromHMM categories ('TSS', 'flanking promoter' and 'bivalent promoter') were filtered out and the analysis described above was repeated. Of the 76,630 peaks retained, 76,221 overlapped at least one chromHMM annotation. We recomputed these correlations (with promoters excluded) after splitting peaks into those less than 30 kilobases from the nearest TSS and peaks greater than or equal to 30 kilobases from the nearest TSS. We also recomputed the correlations for cell type-specifically and ubiquitously expressed genes separately.

## Testing for influence of number of SNVs on p-values and log fold-change

To assign SNVs to genes, we converted all exons in the human gtf file to bed format and intersected the exons with our list of confident human-chimpanzee SNVs. We then used a custom python script to count the number of unique SNVs assigned to each gene and plotted the relationship between DESeq2 p-value and log fold-change estimate. For the ATAC peaks, we performed an identical procedure using the bed file of peaks rather than a bed file of exons. We then plotted the number of SNVs in a peak with the binomial test p-value and raw log fold-change estimate.

## Computation of differential expression enrichment (dEE) and differential chromatin accessibility enrichment (dCAE)

For each target cell type, taking CM as an example, the $\log_2$ fold-change for gene A was fixed as target $\log_2$ fold-change, and the $\log_2$ fold-changes for gene A in the remaining cell types with an opposing sign (compared to the target $\log_2$ fold-change) were set to zero. Then, the dEE value was calculated as the proportion of the target $\log_2$ fold-change in the sum of the zeroed log fold-changes across all cell types. For example, the dEE for gene A in CM would be abs (target LFC)/sum (abs(LFC after zeroing)). dEE ranges from zero to one and low dEE value indicates differential expression with similar magnitude and direction across cell types, and/or the gene does not have any strong allelic bias, whereas a high dEE value indicates that this gene is only strongly differentially expressed (with the sign the $\log_2$ fold-change has in that cell type) in a particular cell type. dCAE uses the same procedure as dEE except the table is populated with the $\log_2$ fold-changes derived from chromatin accessibility measurements. dEE and dCAE are sensitive to the inclusion or exclusion of cell types (by definition), so we excluded RPE when integrative analysis combining results from dEE and dCAE was performed (to match the cell types for which dCAE could be computed, *Figure 3—figure supplement 20b*). After restricting to genes defined as having significant ASE or significant ASCA, we defined genes with dEE ≥ 0.75 in a particular cell type as showing cell-type-specific ASE and peaks with dCAE ≥ 0.75 in one cell type as showing cell-type-specific ASCA. We used bedtools intersect to intersect the peaks

with our list of human-chimpanzee single nucleotide differences and the 241-way placental mammal PhyloP scores (*Quinlan and Hall, 2010*; *Sullivan et al., 2023*). We also checked whether peaks that contained human-chimpanzee differences in sites with high PhyloP scores were in the list of HARs described in *Girskis et al., 2021*.

## Predicting regulatory activity with single-variant resolution

We used sequences in fasta format as input to the deep neural network model Sei (*Chen et al., 2022b*). Sei requires a 4096 base pair input sequence, so we put the center of our region of interest at the center of the input window and expanded equally on either side to contain 4096 base pairs. The human sequence was retrieved from hg38 and the corresponding chimpanzee sequence was retrieved from panTro6. The effect size when comparing the probabilities of each sequence having a particular chromatin state was computed as the log of the human sequence probability divided by the chimpanzee sequence probability. Only annotations for which either the chimpanzee sequence or the human sequence had a probability value greater than or equal to 0.5 were kept for downstream analysis. All SNVs between human and chimpanzee in this input window were identified and ordered based on coordinates. For each SNV position, the human sequence was changed to the chimpanzee allele at that position to generate a new sequence that was input to Sei. The $\log_2$ fold-change for each chromatin annotation was computed for each input sequence as described above and used as a measure of the effect of this change on the sequence. Similarly, an indel can be introduced to modify the human sequence and input to Sei. With indels, the center of the regions of interest (promoter or HAR) were always at the center of the input window and the start or end of the sequence inputted to Sei could possibly lose or gain base pairs. However, we found that for the small indels shown here this had essentially no effect on the Sei output.

## Processing of publicly available datasets

The data from Blake et al. and Pavlovic et al. were processed as previously described (*Blake et al., 2020*; *Pavlovic et al., 2018*; *Starr et al., 2023*). For the Pavlovic et al. data, $\log_2$ fold-changes were computed in DESeq2 with the scaled proportion of cardiomyocytes present in each sample (available in the supplemental materials of Pavlovic et al.), sex, and whether cardiomyocytes were treated with T3 as covariates (i.e. using the model ~sex + scaled_proportion_cardiomyocytes +T3_Treatment +species) (*Pavlovic et al., 2018*). No covariates were included for Blake et al. as they had little impact on the data (*Blake et al., 2020*). The $\log_2$ fold-changes and FDR corrected p-values were directly downloaded from the supplemental materials of *Kozlenkov et al., 2020*.

The processed data from *Ma et al., 2022* were downloaded from http://resources.sestanlab.org/PFC/. We pseudobulked the data by cell type by summing counts within each individual. We then separately input each pairwise comparison of two species (human to chimpanzee or human to rhesus macaque) into DESeq2 with no covariates to test for differential expression and compute $\log_2$ fold-changes.

The counts tables from Kanton et al. were downloaded from https://www.ebi.ac.uk/biostudies/arrayexpress/studies/E-MTAB-7552 and processed with SCANPY (*Kanton et al., 2019*, p. 29409532; *Wolf et al., 2018*). The data were filtered by removing cells with n_genes_by_counts >2500 and >5% mitochondrial reads. We also removed cells with fewer than 200 unique genes and genes that had non-zero counts in fewer than 3 cells. After filtering, any chimpanzee cells not falling in the category (defined by *Kanton et al., 2019*) 'ventral forebrain progenitors and neurons' were eliminated and human cells not in the categories 'ventral progenitors and neurons 1', 'ventral progenitors and neurons 2', or 'ventral progenitors and neurons 3' were similarly eliminated. We then merged the two counts tables, normalized/logarithmized the counts, computed PCA, used harmony (*Korsunsky et al., 2019*) to integrate cells from different species (human and chimpanzee), and found nearest neighbors with the harmonized principal components. We then ran Leiden clustering with resolution = 0.5 to identify 7 subclusters (one of which appeared to be a technical artifact with very low counts that was removed) (*Traag et al., 2019*).

We identified cell types and lineages using canonical marker genes (*MKI67* and *HES5* for progenitors, *NKX2-1* and *LHX6* for the medial ganglionic eminence or MGE, *MEIS2*, and *ZFHX3* for the lateral ganglionic eminence or LGE, and *SCGN* and *NR2F1* for the caudal ganglionic eminence or CGE) (*SuFeher et al., 2022*). We then used the implementation of PAGA in SCANPY to compute

pseudotime using the first cell in the progenitor subcluster as the root (*Wolf et al., 2019*). We binned cells into five equal bins along pseudotime and compared the expression of cells with non-zero counts for *GAD1* in each pseudotime bin. Within each bin, we used a Wilcoxon test to test for higher expression of *GAD1* in chimpanzee cells compared to human cells. We repeated the pseudotime analysis, binning, and comparing of *GAD1* gene expression for each subtrajectory (MGE, LGE, and CGE).

## Acknowledgements

The authors wish to acknowledge the Columbia Stem Cell Core for their hard work in differentiating these cells. We also acknowledge members of the Fraser lab past and present for helpful discussion. Figures 1a, 2a and 3a were created with https://www.biorender.com/. Funding for this work came from NIH R01HG012285. ALS was supported by an NDSEG fellowship.

## Additional information

### Funding

| Funder | Grant reference number | Author |
|---|---|---|
| National Human Genome Research Institute | R01HG012285 | Hunter B Fraser |
| U.S. Department of Defense | NDSEG Graduate Fellowship | Alexander L Starr |

The funders had no role in study design, data collection and interpretation, or the decision to submit the work for publication.

### Author contributions

Ban Wang, Data curation, Software, Formal analysis, Investigation, Visualization, Methodology, Writing – review and editing; Alexander L Starr, Data curation, Software, Formal analysis, Investigation, Visualization, Methodology, Writing – original draft, Writing – review and editing; Hunter B Fraser, Conceptualization, Supervision, Funding acquisition, Writing – review and editing

### Author ORCIDs

Ban Wang ⬛ https://orcid.org/0000-0003-2907-6132
Alexander L Starr ⬛ https://orcid.org/0000-0001-6297-9742
Hunter B Fraser ⬛ https://orcid.org/0000-0001-8400-8541

Reviewer #1 (Public Review): https://doi.org/10.7554/eLife.89594.3.sa1
Reviewer #3 (Public Review): https://doi.org/10.7554/eLife.89594.3.sa2
Author Response https://doi.org/10.7554/eLife.89594.3.sa3

## Additional files

### Supplementary files

• Supplementary file 1. Results of running the test for selection described by *Starr et al., 2023* on each cell type.

• Supplementary file 2. Peak-gene pairs that had concordant high dCAE and high dEE in each cell type.

• MDAR checklist

### Data availability

Raw and processed data generated by this study are publicly available through the Gene Expression Omnibus under accession GSE232949: https://www.ncbi.nlm.nih.gov/geo/query/acc.cgi?acc=GSE232949. The snRNAseq data from Ma et al. are available here: https://resources.sestanlab.org/PFC/. The scRNAseq data from Kanton et al. are available here: https://www.ebi.ac.uk/biostudies/

arrayexpress/studies/E-MTAB-7552. The bulk RNA-seq data from Blake et al. and Pavlovic et al. are available at https://www.ncbi.nlm.nih.gov/geo/query/acc.cgi?acc=GSE112356 and https://www.ncbi.nlm.nih.gov/geo/query/acc.cgi?acc=GSE110471 respectively. The log fold-changes and associated statistics were used directly from the supplemental materials of *Kozlenkov et al., 2020*. The human-chimp pairwise alignment used to identify SNVs and indels is available here: https://hgdownload.soe.ucsc.edu/goldenPath/hg38/vsPanTro6/. The human and chimpanzee genomes used are available here: https://hgdownload.soe.ucsc.edu/goldenPath/hg38/bigZips/ and here: https://hgdownload.soe.ucsc.edu/goldenPath/panTro6/bigZips/ respectively. The 241-way PhyloP scores were downloaded from here: https://hgdownload.soe.ucsc.edu/goldenPath/hg38/cactus241way/. The "QCed genomic constraint by 1kb regions" from the gnomAD consortium are available here: https://gnomad.broadinstitute.org/downloads#v3. All scripts for performing analyses and making figures in this manuscript are publicly available at https://github.com/banwang27/multi-celltypes (copy archived at *Wang and Starr, 2024*).

The following dataset was generated:

| Author(s) | Year | Dataset title | Dataset URL | Database and Identifier |
|---|---|---|---|---|
| Wang B, Starr AL, Fraser HB | 2023 | Cell type-specific cis-regulatory divergence in gene expression and chromatin accessibility revealed by human-chimpanzee hybrid cells | https://www.ncbi.nlm.nih.gov/geo/query/acc.cgi?acc=GSE232949 | NCBI Gene Expression Omnibus, GSE232949 |

The following previously published datasets were used:

| Author(s) | Year | Dataset title | Dataset URL | Database and Identifier |
|---|---|---|---|---|
| Camp G, Treutlein B, He Z, Kanton S | 2019 | Single cell RNA-seq of great ape cerebral organoids | https://www.ebi.ac.uk/biostudies/arrayexpress/studies/E-MTAB-7552 | ArrayExpress, E-MTAB-7552 |
| Roux J, Blake LE, Hernando-Herraez I, Hsiao CJ, Banovich NE, Garcia Perez R, Chavarria C, Mitrano A, Pritchard JK, Marques-Bonet T, Gilad Y | 2018 | A genomic study of the contribution of DNA methylation to regulatory evolution in primates | https://www.ncbi.nlm.nih.gov/geo/query/acc.cgi?acc=GSE112356 | NCBI Gene Expression Omnibus, GSE112356 |
| Pavlovic BJ, Blake LE, Chavarria C, Gilad Y | 2018 | A Comparative Assessment of iPSC Derived Cardiomyocytes with Heart Tissues in Humans and Chimpanzees | https://www.ncbi.nlm.nih.gov/geo/query/acc.cgi?acc=GSE110471 | NCBI Gene Expression Omnibus, GSE110471 |

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
