## [Editor Report · eLife assessment]

This is an **important** study that leverages a human-chimpanzee tetraploid iPSC model to test whether *cis*-regulatory divergence between species tends to be cell type-specific. The evidence supporting the study's primary conclusions together provide **convincing** evidence for enrichment of species differences in gene regulation in cell type-specific genes and regulatory elements, motivating future work with larger sample sizes of cell lines. This work will be of broad interest in evolutionary and functional genomics.

---

## [Referee Report · Reviewer #1 (Public Review)]

This study aims to identify gene expression differences exclusively caused by *cis*-regulatory genetic changes by utilizing hybrid cell lines derived from human and chimpanzee. While previous attempts have focused on specific tissues, this study expands the comparison to six different tissues to investigate tissue specificity and derive insights into the evolution of gene expression.

One notable strength of this work lies in the use of composite cell lines, enabling a comparison of gene expression between human and chimpanzee within the same nucleus and shared trans factors environment. However, a potential weakness of the methodology is the use of bulk RNA-seq in diverse tissues, which limits the ability to determine cell-type-specific gene expression and chromatin accessibility regions. Their approach, using hybrid lines, naturally accounts for cell type heterogeneity avoiding the risk of false positives introduced by the otherwise confounding differences in cell type abundances between species, albeit the challenge of false negatives remains an issue. The authors now dully acknowledge this limitation in the manuscript.

Another concern is the use of two replicates derived from the same pair of individuals. While the authors produced cell lines from two pairs of individuals in a previous study (Agloglia et al., 2021). The reason for this experimental design is cost limitations. The authors now acknowledge that the use of replicates could enhance the ability to detect "more" species-specific changes in expression and chromatin accessibility. I would emphasize that replicates would increase robustness to the present findings, given that they are derived from a single pair of individuals.

Furthermore, the study offers the opportunity to relate inter-species differences to trends in molecular evolution. The authors discovered that expression variance and haploinsufficiency score do not fully account for the enrichment of divergence in cell-type-specific genes. The reviewer suggested exploring this further by incorporating external datasets that bin genes based on interindividual transcriptomics variation as a measure of extant transcriptomics constraint (e.g., GTEx reanalysis by Garcia-Perez et al., 2023 -- PMID: 36777183). The authors considered this question to be out of the scope of the paper, yet in my opinion this would enhance one of the main findings of this study.

Additionally, stratifying sequence conservation on ASCA regions, which exhibit similar enrichment of cell-type-specific features, using the Zoonomia data mentioned also in the text (Andrews et al., 2023 -- PMID: 37104580) could provide valuable insights. While the author did not find Zoonomia Phastcons values available, they used PhastCons derived from a 470-way alignment of mammals. I commend the authors for their diligent efforts, which undoubtedly bolster their findings that an enrichment in ASCA is evident across all levels of sequence conservation. However, this recent analysis indicates the presence of a potential relationship between sequence conservation and ASCA. It may be advantageous to consider evaluating more quantile subdivisions of maxZ values and pPhastCons values, with the inclusion of these results in the supplementary materials. This approach would be preferable, even if the precise reasons behind the observed discrepancy are not fully elucidated.

Another potential strength of this study is the identification of specific cases of paired allele-specific expression (ASE) and allele-specific chromatin accessibility (ASCA) with biological significance. Prioritizing specific variants remains a challenge, and the authors apply a machine learning approach to identify potential causative variants that disrupt binding sites in two examples (FABP7 and GAD1 in motor neurons). However, additional work is needed to convincingly demonstrate the functionality of these selected variants. Strengthening this section with additional validation of ASE, ASCA, and the specific putative causal variants identified would enhance the overall robustness of the paper. The authors have opted to defer these validations to future studies.

Additionally, the authors support the selected ASE-ASCA pairs by examining external datasets of adult brain comparative genomics (Ma et al., 2022) and organoids (Kanton et al., 2019). While these resources are valuable for comparing observed species biases, the analysis is not systematic, even for the two selected genes. For example, it would be beneficial to investigate if FABP7 exhibits species bias in any cell type in Kanton et al.'s organoids or if GAD1 is species-biased in adult primate brains from Ma et al. Comparing these datasets with the present study, along with the Agoglia et al. reference, would provide a more comprehensive perspective. In the revised version of the manuscript the authors have evaluated the expression of GAD1 in Ma et al, and FABP7 in Sousa et al 2017. For instance, GAD1 show cell type specific species biases in the later. The authors opted for not showing this in the manuscript, However, it remains unclear why certain datasets were favored over others, or why FABP7 should not be evaluated in Kanton et al.

The use of the term "human-derived" in ASE and ASCA has now been avoided.

Finally, throughout the paper, the authors refer to "hybrid cell lines." It has been suggested to use the term "composite cell lines" instead to address potential societal concerns associated with the term "hybrid," which some may associate with reproductive relationships (Pavlovic et al., 2022 -- PMID: 35082442). The authors have presented an eloquent and persuasive explanation that I found to be highly informative.

---

## [Referee Report · Reviewer #3 (Public Review)]

The authors utilize chimpanzee-human hybrid cell lines to assess *cis*-regulatory evolution. These hybrid cell lines offer a well-controlled environment, enabling clear differentiation between *cis*-regulatory effects and environmental or other trans effects.

In their research, Wang et al. expand the range of chimpanzee-human hybrid cell lines to encompass six new developmental cell types derived from all three germ layers. This expansion allows them to discern cell type-specific *cis*-regulatory changes between species from more pleiotropic ones. Although the study investigates only two iPSC clones, the RNA- and ATAC-seq data produced for this paper is a valuable resource.

The authors begin their analysis by examining the relationship between allele-specific expression (ASE) as a measure of species divergence and cell type specificity. They find that cell-type-specific genes exhibit more divergent expression. By integrating this data with measures of constraint within human populations, the authors conclude that the increased divergence of tissue-specific genes is, at least in part, attributable to positive selection. A similar pattern emerges when assessing allele-specific chromatin accessibility (ASCA) as a measure of divergence of *cis*-regulatory elements (CREs) in the same cell lines.

By correlating these two measures, the authors identify 95 CRE-gene pairs where tissue-specific ASE aligns with tissue-specific ASCA. Among these pairs, the authors select two genes of interest for further investigation. Notably, the authors employ an intriguing machine learning approach in which they compare the inferred chromatin state of the human sequence with that of the chimpanzee sequence to pinpoint putatively causal variants.

Overall, this study delves into the examination of gene expression and chromatin accessibility within hybrid cell lines, showcasing how this data can be leveraged to identify potential causal sequence differences underlying between-species expression changes.

All in all most conclusions appear solid, with the exception of the interpretation of a cell type/state identification machine learning model to pinpoint putatively causal variants. The described variants lack any functional validation and there is no data that measure the certainty of the results.

---

## [Author Response]

The following is the authors’ response to the original reviews.

**eLife assessment**
This is an important study that leverages a human-chimpanzee tetraploid iPSC model to test whether *cis*-regulatory divergence between species tends to be cell type-specific. The evidence supporting the study's primary conclusion--that species differences in gene regulation are enriched in cell type-specific genes and regulatory elements--is compelling, although attention to biases introduced by sequence conservation is merited, and the case that is made for cell type-specific changes reflecting adaptive evolution is incomplete. This work will be of broad interest in evolutionary and functional genomics.
**Public Reviews:**

**Reviewer #1 (Public Review):**
This study aims to identify gene expression differences exclusively caused by *cis*-regulatory genetic changes by utilizing hybrid cell lines derived from human and chimpanzee. While previous attempts have focused on specific tissues, this study expands the comparison to six different tissues to investigate tissue specificity and derive insights into the evolution of gene expression.One notable strength of this work lies in the use of composite cell lines, enabling a comparison of gene expression between human and chimpanzee within the same nucleus and shared trans factors environment. However, a potential weakness of the methodology is the use of bulk RNA-seq in diverse tissues, which limits the ability to determine cell-type-specific gene expression and chromatin accessibility regions.

We agree that profiling single cells could lead to additional exciting discoveries. Although heterogeneity in cell types within samples will indeed reduce our power to detect cell-type-specific divergence, thankfully any heterogeneity will not introduce false positives, since our use of interspecies hybrids controls for differences in cell-type abundance. As a result, we think that the molecular differences we identified in this study represent a subset of the true cell-type-specific *cis*-regulatory differences that would be identified with deep single-cell profiling. We have included a new paragraph in the discussion on future directions, highlighting the utility of single-cell profiling as an exciting future direction (lines 482-490): “In addition to following up on our findings on GAD1 and FABP7, there are other exciting future directions for this work. First, additional bulk assays such as those that measure methylation, chromatin conformation, and translation rate could lead to a better understanding of what molecular features ultimately lead to cell type-specific changes in gene expression. Furthermore, the use of deep single cell profiling of hybrid lines derived from iPSCs from multiple individuals of each species during differentiation could enable the identification of many more highly context-specific changes in gene expression and chromatin accessibility such as the differences in GAD1 we highlighted here. Finally, integration with data from massively parallel reporter assays and deep learning models will help us link specific variants to the molecular differences we identified in this study.”

Another concern is the use of two replicates derived from the same pair of individuals. While the authors produced cell lines from two pairs of individuals in a previous study (Agloglia et al., 2021), I wonder why only one pair was used in this study. Incorporating interindividual variation would enhance the robustness of the species differences identified here.

We agree that additional replicates, especially from lines from other individuals, would have improved the robustness of the species differences we identified. In our experience with these hybrid cells (as well as related work from many other labs), inter-species differences typically have much larger magnitudes than intra-species differences, so we expect that the vast majority of differences we identified would be validated with data from additional individuals. Unfortunately, differentiating additional cells and generating these data for this study would be cost-prohibitive. We now mention the use of additional replicates in lines 485-488 of the discussion: “Furthermore, the use of deep single cell profiling of hybrid lines derived from iPSCs from multiple individuals of each species during differentiation could enable the identification of many more highly context-specific changes in gene expression and chromatin accessibility such as the differences in GAD1 we highlighted here.”

Furthermore, the study offers the opportunity to relate inter-species differences to trends in molecular evolution. The authors discovered that expression variance and haploinsufficiency score do not fully account for the enrichment of divergence in cell-type-specific genes. The reviewer suggests exploring this further by incorporating external datasets that bin genes based on interindividual transcriptomics variation as a measure of extant transcriptomics constraint (e.g., GTEx reanalysis by Garcia-Perez et al., 2023 -- PMID: 36777183). Additionally, stratifying sequence conservation on ASCA regions, which exhibit similar enrichment of cell-type-specific features, using the Zoonomia data mentioned also in the text (Andrews et al., 2023 -- PMID: 37104580) could provide valuable insights.

To address this, we used PhastCons scores computed from a 470-way alignment of mammals as we could not find publicly available PhastCons data from Zoonomia. When stratifying by the median PhastCons score of all sites in a peak, we observe very similar results to those obtained when stratifying by the constraint metric from the gnomAD consortium (see below). The one potential difference is that peaks in the top two bins have slightly weaker enrichment relative to the other bins when using PhastCons, but this is not the case when using gnomAD’s metric. We have elected to include this in the public review but not the manuscript as we are reluctant to add to the complexity of what is already complex analysis.

Finally, we think that comparisons of the properties of gene expression variance computed from ASE (as done by Starr et al.) and total expression (as done by Garcia-Perez et al.) is a very interesting, potentially complex question that is beyond the scope of this paper but an exciting direction for future work.

Another potential strength of this study is the identification of specific cases of paired allele-specific expression (ASE) and allele-specific chromatin accessibility (ASCA) with biological significance.Prioritizing specific variants remains a challenge, and the authors apply a machine-learning approach to identify potential causative variants that disrupt binding sites in two examples (FABP7 and GAD1 in motor neurons). However, additional work is needed to convincingly demonstrate the functionality of these selected variants. Strengthening this section with additional validation of ASE, ASCA, and the specific putative causal variants identified would enhance the overall robustness of the paper.

We strongly agree with the reviewer that additional work validating our results would be of considerable interest. We hope to perform follow-up experiments in the future. For now, we have been careful to present these variants only as candidate causal variants.

Additionally, the authors support the selected ASE-ASCA pairs by examining external datasets of adult brain comparative genomics (Ma et al., 2022) and organoids (Kanton et al., 2019). While these resources are valuable for comparing observed species biases, the analysis is not systematic, even for the two selected genes. For example, it would be beneficial to investigate if FABP7 exhibits species bias in any cell type in Kanton et al.'s organoids or if GAD1 is species-biased in adult primate brains from Ma et al. Comparing these datasets with the present study, along with the Agoglia et al. reference, would provide a more comprehensive perspective.

We agree with the reviewer’s suggestion that investigating GAD1 and FABP7 expression in other datasets is worthwhile. Unfortunately, the difference in human vs. chimpanzee organoid maturation rates and effects of culture conditions in Kanton et al. makes it unsuitable for plotting the expression of FABP7 as its expression is highly dependent on neuronal maturation. We therefore plotted bulk RNAseq data from multiple cortical regions from Sousa et al. 2017 (see below). This corroborates our claim that FABP7 has human-biased expression in adult humans compared to chimpanzees and rhesus macaques. We also investigated expression of GAD1 in the Ma et al. data as the reviewer suggested.

**Author response image 2. sa3fig2:** 

While there are differences in GAD1 expression between adult humans and chimpanzees, they are unlikely to be linked to the HAR we highlight as it is likely a transiently active *cis*-regulatory element (see below). In addition, some cell types seem to have chimpanzee-derived changes in GAD1 expression (e.g. SST positive neurons) whereas others seem to have human-derived changes in GAD1 expression (e.g. LAMP5 positive neurons).

**Author response image 3. sa3fig3:** 

While these are potentially interesting observations, we think that their inclusion in the manuscript might distract from our emphasis on the cell type-specific and developmental stage-specific of the changes in FABP7 and GAD1 expression we observe so we have not included them in the manuscript.

The use of the term "human-derived" in ASE and ASCA should be avoided since there is no outgroup in the analysis to provide a reference for the observed changes.

We agree with the reviewer that the term human-derived should be used with care and have changed the phrasing of line 230 to “human-chimpanzee differences in expression”. With regard to FABP7 we think that our analysis of the Ma et al. data—which includes data from rhesus macaques as an outgroup—justifies our use of “human-derived” in lines 360 and 457. As chimpanzee and macaque expression of FABP7 are similar but human expression is quite different, the most parsimonious explanation for our observations is that FABP7 upregulation occurred in the human lineage.

Finally, throughout the paper, the authors refer to "hybrid cell lines." It has been suggested to use the term "composite cell lines" instead to address potential societal concerns associated with the term "hybrid," which some may associate with reproductive relationships (Pavlovic et al., 2022 -- PMID: 35082442). It would be interesting to know the authors' perspective on these concerns and recommendations presented in Pavlovic et al., given their position as pioneers in this field.

We appreciate this question. Whether to refer to our fused cells as “hybrids” or not was indeed a question we considered at great length, starting from the very beginning of this project in 2015. From consultations with multiple bioethicists-- both formal and informal-- we have long been aware of the possibility of misunderstanding based on the word “hybrid”. However, we felt this possibility was outweighed by the long and well-established history of other scientists referring to interspecies fused cells as hybrids. This convention-- which is based on hundreds of papers about heterokaryons, somatic cell hybrids, and radiation hybrids-- goes back over 50 years (e.g. Bolund et al, Exp Cell Res 1969). Soon after the establishment of this nomenclature, cell fusion became widespread and ever since then it has become commonplace to generate interspecies hybrid cells from animals, plants and fungi.

It is also important to note that in over two years since we published the first two papers on human/chimpanzee fused cells, we have been unable to find any misunderstanding of our use of the term “hybrid”. We have searched blogs, media articles, and social media, all with no evidence of misunderstanding. Therefore, in the current manuscript, rather than creating confusion by renaming a well-established approach, we have opted to clearly and prominently define hybrid cells: in the abstract of our paper we introduce the hybrid cells as “the product of fusing induced pluripotent stem (iPS) cells of each species *in vitro*.”

**Reviewer #2 (Public Review):**
In this paper, Wang and colleagues build on previous technical and analytical achievements in establishing tetraploid human-chimpanzee hybrid iPSCs to investigate the cell type-specificity of allelespecific expression and allele-specific chromatin accessibility across six differentiated cell types (here, "allele-specific" indicates species differences with a *cis*-regulatory basis). The combined body of work is remarkable in its creativity and ambition and has real potential for overcoming major challenges in understanding the evolutionary genetics of between-species differences. The present paper contributes to these efforts by showing how differentiated cells can be used to test a long-standing hypothesis in evolutionary genetics: that *cis*-regulatory changes may be particularly important in divergence because of their potential for modularity.In my view, the paper succeeds in making this case: allele (species)-specific expression (ASE) and allelespecific chromatin accessibility (ASCA) are enriched in genes asymmetrically expressed in one cell type, and many cases of ASE/ASCA are cell type-specific. The authors do an excellent job showing that these results are robust across a set of possible analysis decisions. It is somewhat less clear whether these enrichments are primarily a product of relaxed constraint on cell type-specific genes or primarily result from positive selection in the human or chimp lineage. While the authors attempt to control for constraint using several variables (variance in ASE in humans and the sequence-based probability of haploinsufficiency score, pHI), these are imperfect proxies for constraint. For the pHI scores, enrichments for ASE also appear to be strongest in the least constrained genes. Overall, the relative role of relaxation of constraint versus positive selection is unresolved, although the manuscript's language leans in favor of an important role for selection.

We agree with the reviewer and apologize for the wording that indeed focused more on positive selection than relaxed constraint. We have added language clarifying that our stance is that our analyses suggest some role for positive selection, but that we do not claim that positive selection plays a larger role than reduced constraint (lines 432-437): “Overall, this suggests that broad changes in expression in cell type-specifically expressed genes may be an important substrate for evolution but it remains unclear whether positive selection or lower constraint plays a larger role in driving the faster evolution of more cell type-specifically expressed genes. Future work will be required to more precisely quantify the relative roles of positive selection and evolutionary constraint in driving changes in gene expression.”

The remainder of the manuscript draws on the cell type-specific ASE/ASCA data to nominate candidate genes and pathways that may have been important in differentiating humans and chimpanzees. Several approaches are used here, including comparing human-chimp ASE to the distribution of ASE observed in humans and investigating biases in the direction of ASE for genes in the same pathway. The authors also identify interesting candidate genes based on their role in development or their proximity to human accelerated regions (where many changes have arisen on the human lineage in otherwise deeply conserved sequence) and use a deep neural network to identify sequence changes that might be causally responsible for ASE/ASCA. These analyses have value and highlight potential strategies for using ASE/ASCA and hybrid cell line data as a hypothesis-generating tool. Of course, the functional follow-up that experimentally tested these hypotheses or linked sequence/expression changes in the candidate pathways to organismal phenotype would have strengthened the paper further- but this is a lot to ask in an already technically and analytically challenging piece of work.

We thank the reviewer for the kind words and strongly agree that follow-up experiments and orthogonal analyses will be key in validating our results and establishing links to human-specific phenotypes.

As a minor critique, the present paper is very closely integrated with other manuscripts that have used the hybrid human-chimp cell lines for biological insight or methods development. Although its contributions make it a strong stand-alone contribution, some aspects of the methods are not described in sufficient detail for readers to understand (even on a general conceptual level) without referencing that work, which may somewhat limit reader understanding.

We agree with the points the reviewer raises regarding the clarity of our methods. We have amended several sections to provide more conceptual information while pointing the reader to other publications for the technical details. For convenience, we include the text here as well as in the new draft.

Lines 207-214 now provide more intuition for the method used to detect lineage-specific selection: “Next, we sought to use our RNA-seq data to identify instances of lineage-specific selection. In the absence of positive selection, one would expect that an approximately equal number of genes in a pathway would have human-biased vs. chimpanzee-biased ASE. Significant deviation from this expectation (as determined by the binomial test) rejects the null hypothesis of neutral evolution, instead providing evidence of lineage-specific selection on this pathway. Using our previously published modification of this test that incorporates a tissue-specific measure of constraint on gene expression, we detected several signals of lineage-specific selection, some of which were cell type-specific (Starr et al., 2023, Additional file 2).” This is also reflected in the Methods in lines 729-731: “Positive selection on a gene set is only inferred if there is statistically significant human- or chimpanzee-biased ASE in that gene set (using an FDR-corrected p-value from the binomial test).”

**Reviewer #3 (Public Review):**
The authors utilize chimpanzee-human hybrid cell lines to assess *cis*-regulatory evolution. These hybrid cell lines offer a well-controlled environment, enabling clear differentiation between *cis*-regulatory effects and environmental or other trans effects.In their research, Wang et al. expand the range of chimpanzee-human hybrid cell lines to encompass six new developmental cell types derived from all three germ layers. This expansion allows them to discern cell type-specific *cis*-regulatory changes between species from more pleiotropic ones. Although the study investigates only two iPSC clones, the RNA- and ATAC-seq data produced for this paper is a valuable resource.The authors begin their analysis by examining the relationship between allele-specific expression (ASE) as a measure of species divergence and cell type specificity. They find that cell-type-specific genes exhibit more divergent expression. By integrating this data with measures of constraint within human populations, the authors conclude that the increased divergence of tissue-specific genes is, at least in part, attributable to positive selection. A similar pattern emerges when assessing allele-specific chromatin accessibility (ASCA) as a measure of divergence of *cis*-regulatory elements (CREs) in the same cell lines.By correlating these two measures, the authors identify 95 CRE-gene pairs where tissue-specific ASE aligns with tissue-specific ASCA. Among these pairs, the authors select two genes of interest for further investigation. Notably, the authors employ an intriguing machine-learning approach in which they compare the inferred chromatin state of the human sequence with that of the chimpanzee sequence to pinpoint putatively causal variants.Overall, this study delves into the examination of gene expression and chromatin accessibility within hybrid cell lines, showcasing how this data can be leveraged to identify potential causal sequence differences underlying between-species expression changes.

We appreciate this assessment.

I have three major concerns regarding this study:1. The only evidence that the cells are indeed differentiated in the right direction is the expression of one prominent marker gene per cell type. Especially for the comparison of conservation between the differentiated cell types, it would be beneficial to describe the cell type diversity and the differentiation success in more detail.

We appreciate this assessment. We agree that evidence beyond a single marker gene is necessary to demonstrate that the differentiations were successful and that a discussion of the limitations of these differentiations in the manuscript is worthwhile. We included figures showing additional marker genes and a thorough discussion of the differentiations in the supplement. For convenience, we have copied the supplemental figure and text here:

“Before continuing with the analysis, we tested whether the differentiations were successful and contained primarily our target cell types. The very low expression of NANOG, a marker for pluripotency, across all differentiations indicates that the samples contain very few iPSCs (Agoglia et al., 2021). For cardiomyocytes (CM), NKX2-5, MYBPC3, and TNNT2 definitively distinguish CM from other heart cell types and their high expression indicates successful differentiations (Burridge et al., 2014). For motor neurons, the high expression of ELAVL2, a pan-neuronal marker, indicates a high abundance of neurons in the sample (Mickelsen et al., 2019). The expression of ISL1 and OLIG2 further demonstrates that these are motor neurons and not other types of neurons (Maury et al., 2015). For retinal pigment epithelium(RPE), the combined expression of MITF, PAX6, and TYRP1 provides strong evidence that the differentiations were successful in producing RPE cells (Sharma et al., 2019). For skeletal muscle, the very high expression of MYL1, MYLPF, and MYOG indicates that these samples contain a high proportion of skeletal muscle cells (Chal et al., 2016). In general, all these populations of cells contain some proportion of progenitors as there is detectable expression of MKI67 in all samples.

The low expression of ALB (a marker for mature hepatocytes) and the high expression of TTR and GPC3 (markers for hepatocyte progenitors) combined with the high expression of HNF1B indicate that the bulk of the cells in the HP samples are hepatocyte progenitors rather than mature hepatocytes or endoderm cells, although there are likely some endoderm cells and immature hepatocytes in the sample (Hay et al., 2008; Mallanna & Duncan, 2013). Similarly, the combined expression of PDX1 and NKX6-1 and the low expression of NEUROG3 (a marker of endocrine progenitors which differentiate from pancreatic progenitors) in the PP samples indicates that these primarily contain pancreatic progenitors but likely contain some endocrine progenitors and endoderm cells (Cogger et al., 2017; Korytnikov & Nostro, 2016).

Notably, HP and PP are closely related cell types that are derived from the same lineage. Indeed, heterogeneous multipotent progenitors can contribute to both the adult liver and adult pancreas in mice (Willnow et al., 2021). Progenitors that express PDX1 (often used as a marker for the pancreatic lineage) can differentiate into hepatocytes (Willnow et al., 2021). As a result, some overlap in the transcriptomic signature of both cell types is expected and we cannot rule out that the HP samples contain cells that could differentiate into pancreatic cells or that the PP samples contain cells that could differentiate into hepatocytes. However, the expression of NKX6-1 and GP2, markers for pancreatic progenitors, in the PP samples but not the HP samples indicates that these two populations of cells are distinct. Overall, the similarity of PP and HP likely explains the lower number of cell type-specific genes and genes showing cell type-specific ASE for these cell types. This similarity does not alter the conclusions presented in the main text.”

**Author response image 4. sa3fig4:** 

**Author response image 5. sa3fig5:** Marker gene expression in different cell types. In order, the panels show: a marker for pluripotency, a marker gene for dividing cells, marker genes for cardiomyocytes, marker genes for hepatocytes and hepatocyte progenitors, marker genes for motor neurons, marker genes for pancreatic progenitors and more mature pancreatic cell types, marker genes for retinal pigment epithelial cells, and marker genes for skeletal myocytes. Hepatocyte progenitors and pancreatic progenitors generally show similar gene expression profiles. TPM: transcript per million.

1. Check for a potential confounding effect of sequence similarity on the power to detect ASE or ASCA.

We agree that checking for confounding by power to detect ASE or ASCA would increase confidence in our results. We have added supplementary figures 29-33 to show the results as well as a discussion of these figures in the text (lines 318-326):

“Finally, it is possible that CREs and genes that are less conserved will have more SNPs, and therefore more power to call ASCA and ASE, leading to systematically biased estimates. There is a weak positive correlation between the number of SNPs and the -log_10_(FDR) for ASE and a weak negative or no correlation for ASCA (Supp Fig. 29). Similarly, we observe a weak relationship between the number of SNPs in CREs or genes and absolute log fold-change estimates (Supp Fig. 30). Although the relationship between the number of SNPs and ASE/ASCA is weak, we confirmed that cell type-specific genes and peaks are still strongly enriched for ASE and ASCA when stratifying by number of SNPs (Supp Fig. 31-32). Overall, our analysis suggests that the result that more cell type-specific genes and CREs are more evolutionarily diverged is robust to a variety of possible confounders.”

**Author response image 6. sa3fig6:** Relationship between number of SNPs and -log_10_(FDR) in (a) ASE and -log_10_(pvalue) (b) ASCA. These scatter plots show the relationship between the number of SNPs in a gene or peak and the -log_10_(FDR) for ASE or ASCA. Genes with significant ASE (FDR < 0.05) and peaks with significant ASCA (binomial p-value < 0.05) were annotated as blue dots, and all other genes and peaks were annotated as gray dots. All genes in each cell type in RNA-seq are shown. For clarity, the few outlier peaks with more than 200 SNPs are excluded from these plots.

**Author response image 7. sa3fig7:** Relationship between number of SNPs and absolute log_2_ fold-change in (a) ASE and (b) ASCA. These scatter plots show the relationship between the number of SNPs in a gene or peak and the estimated absolute log_2_ fold-change for ASE or ASCA. Genes with significant ASE (FDR < 0.05) and peaks with significant ASCA (binomial p-value < 0.05) were annotated as blue dots, and all other genes and peaks were annotated as gray dots. All genes in each cell type in RNA-seq are shown. For clarity, the few outlier peaks with more than 200 SNPs are excluded from these plots.

**Author response image 8. sa3fig8:** Cell type-specifically expressed genes are enriched for genes with ASE when stratifying by the number of SNPs per gene. a) Results when SKM is included. Genes were put into five bins with an equal number of genes in each bin. Genes with the fewest SNPs are in the 0-20% bin and genes with the most SNPs are in the 80-100% bin. Significance (using the Wald test) is indicated by asterisks where *** indicates p < 0.005, ** indicates p < 0.01, and * indicates p < 0.05. b) The same as in (a) but excluding SKM.

**Author response image 9. sa3fig9:** Cell type-specific peaks are enriched for ASCA when stratifying by the number of SNPs per peak. a) Peaks with an absolute log_2_ fold-change greater than or equal to 0.5 were called as having ASCA. Peaks were put into five bins with an equal number of peaks in each bin. Peaks with the fewest SNPs are in the 0-20% bin and genes with the most SNPs are in the 80-100% bin. Significance (using the Wald test) is indicated by asterisks where *** indicates p < 0.005, ** indicates p < 0.01, and * indicates p < 0.05. b) The same as in (a) but peaks with a binomial p-value less than or equal to 0.05 were called as having ASCA.

1. In the last part the authors showcase 2 examples for which the log_2_ fold changes in chromatin state scores as inferred by the machine learning model Sei are used. This is an interesting and creative approach, however, more sanity checks on this application are necessary.

We agree with the reviewer about the importance of sanity checks and apologize for omitting these from the manuscript. Below we highlight several such checks from previous publications:

In the original Sei paper (Chen et al. 2022), the authors included several tests of their model’s ability to predict the effects on individual genetic variants. Using eQTL data from GTEx, they found that variants predicted to increase enhancer activity were more likely to be up-regulating eQTLs, and those predicted to increase polycomb repression had the expected repressive effect. These relationships became stronger when restricting the analysis only to fine-mapped eQTLs with >95% posterior probabilities of causality. Chen et al. also found that previously known disease-causing noncoding variants from the Human Gene Mutation Database were far more likely to reduce predicted enhancer/promoter activity than matched variants not linked to any disease.

In addition, we note that a similar approach to ours was recently used to analyze all HARs and included considerable efforts to validate the utility of the Sei predictions in identifying causal variants (Whalen et al. 2023 in Neuron). For example, Whalen et al. found that the Sei output correlated with the effects of genetic variants on expression in a massively parallel reporter assay. They also found that the effect sizes predicted by Sei were much higher for variants in HARs than polymorphic variants in the human population, which is consistent with the idea that variants in HARs lie in highly conserved bases that are more likely to disrupt *cis*-regulatory elements. Finally, Whalen et al. found that effects on chromatin state predicted by Sei were generally highly correlated across tissues, supporting our approach that leverages all Sei outputs regardless of which cell type or tissue they correspond to. Overall, we think that Sei is a potentially powerful way to prioritize causal variants and that improved machine learning models trained on more extensive and context-specific data will be even more powerful.